# Towards a Theoretical Framework of Out-of-Distribution Generalization

**Haotian Ye**
Peking University
Pazhou Lab
haotianye@pku.edu.cn

**Chuanlong Xie**
Huawei Noah's Ark Lab
xie.chuanlong@huawei.com

**Tianle Cai**
Peking University
caitianle1998@pku.edu.cn

**Ruichen Li**
Peking University
xk-lrc@pku.edu.cn

**Zhenguo Li**
Huawei Noah's Ark Lab
Li.Zhenguo@huawei.com

**Liwei Wang**
Key Laboratory of Machine Perception,
MOE, School of EECS,
Institute for Artificial Intelligence,
Peking University
wanglw@cis.pku.edu.cn

## Abstract

Generalization to out-of-distribution (OOD) data is one of the central problems in modern machine learning. Recently, there is a surge of attempts to propose algorithms that mainly build upon the idea of extracting invariant features. Although intuitively reasonable, theoretical understanding of what kind of invariance can guarantee OOD generalization is still limited, and generalization to arbitrary out-of-distribution is clearly impossible. In this work, we take the first step towards rigorous and quantitative definitions of 1) what is OOD; and 2) what does it mean by saying an OOD problem is learnable. We also introduce a new concept of expansion function, which characterizes to what extent the variance is amplified in the test domains over the training domains, and therefore give a quantitative meaning of invariant features. Based on these, we prove OOD generalization error bounds. It turns out that OOD generalization largely depends on the expansion function. As recently pointed out by [21], any OOD learning algorithm without a model selection module is incomplete. Our theory naturally induces a model selection criterion. Extensive experiments on benchmark OOD datasets demonstrate that our model selection criterion has a significant advantage over baselines.

## 1   Introduction

One of the most fundamental assumptions of classic supervised learning is the "*i.i.d.* assumption", which states that the training and the test data are independent and identically distributed. However, this assumption can be easily violated in a reality [8, 10, 11, 17, 38, 48, 56] where the test data usually have a different distribution than the training data. This motivates the research on the out-of-distribution (OOD) generalization, or domain generalization problem, which assumes access only to data drawn from a set $\mathcal{E}_{avail}$ of available domains during training, and the goal is to generalize to a larger domain set $\mathcal{E}_{all}$ including *unseen* domains.

To generalize to OOD data, most existing algorithms attempt to learn features that are *invariant* to a certain extent across training domains in the hope that such invariance also holds in unseen domains. For example, distributional matching-based methods [20, 35, 55] seek to learn features that have the same distribution across different domains; IRM [5] and its variants [1, 32, 33] learn feature representations such that the optimal linear classifier on top of the representation matches across domains.

35th Conference on Neural Information Processing Systems (NeurIPS 2021).

Though the idea of learning invariant features is intuitively reasonable, there is only limited theoretical understanding of what kind of invariance can guarantee OOD generalization. Clearly, generalization to an arbitrary out-of-distribution domain is impossible and in practice, the features can hardly be absolutely invariant from $\mathcal{E}_{avail}$ to $\mathcal{E}_{all}$ unless all the domains are identical. So it is necessary to first formulate what OOD data can be generalized to, or, what is the relation between the available training domain set $\mathcal{E}_{avail}$ and the entire domain set $\mathcal{E}_{all}$. Meanwhile, to what extent the invariance of features on $\mathcal{E}_{avail}$ can be preserved in $\mathcal{E}_{all}$ should be rigorously characterized.

In this paper, we take the first step towards a general OOD framework by quantitatively formalizing the relationship between $\mathcal{E}_{avail}$ and $\mathcal{E}_{all}$ in terms of the distributions of *features* and provide OOD generalization guarantees based on our quantification of the difficulty of OOD generalization problem. Specifically, we first rigorously formulate the intuition of invariant features used in previous works by introducing the "variation" and "informativeness" (Definition 3.1 and 3.2) of each feature. Our theoretical insight can then be informally stated as: for learnable OOD problems, if a feature is informative for the classification task as well as invariant over $\mathcal{E}_{avail}$, then it is still invariant over $\mathcal{E}_{all}$. In other words, invariance of informative features in $\mathcal{E}_{avail}$ can be preserved in $\mathcal{E}_{all}$. We further introduce a class of functions, dubbed expansion function (Definition 3.3), to quantitatively characterize to what extent the variance of features on $\mathcal{E}_{avail}$ is amplified on $\mathcal{E}_{all}$.

Based on our formulation, we derive theoretical guarantees on the OOD generalization error, i.e., the gap of largest error between the domain in $\mathcal{E}_{avail}$ and domain in $\mathcal{E}_{all}$. Specifically, we prove the upper and lower bound of OOD generalization error in terms of the expansion function and the variation of learned features over $\mathcal{E}_{avail}$. Our results theoretically confirm that 1) the expansion function can reflect the difficulty of OOD generalization problem, i.e., problems with more rapidly increasing expansion functions are harder and have worse generalization guarantees; 2) the generalization error gap can tend to zero when the variation of learned features tend to zero, so minimizing the variation in $\mathcal{E}_{avail}$ can reduce the generalization error.

As pointed out by Gulrajani and Lopez-Paz [21], any OOD algorithm without a specified model selection criterion is not complete. Since $\mathcal{E}_{all}$ is unseen, hyper-parameters can only be chosen according to $\mathcal{E}_{avail}$. Previous selection methods mainly focus on validation accuracy over $\mathcal{E}_{avail}$, which is only a biased metric of OOD performance. On the contrary, a promising model selection method should instead be predictive of OOD performance. Inspired by our bounds, we propose a model selection method to select models with high validation accuracy and low variation, which corresponds to the upper bound of OOD error. The introduction of a model's variation relieves the problem of classic selection methods, in which models that overfit $\mathcal{E}_{avail}$ tend to be selected. Experimental results show that our method can outperform baselines and select models with higher OOD accuracy.

**Contributions.** We summarize our major contributions here:

- We introduce a quantitative and rigorous formulation of OOD generalization problem that characterizes the relation of invariance over the training domain set $\mathcal{E}_{avail}$ and test domain set $\mathcal{E}_{all}$. The core quantity in our characterization, the expansion function, determines the difficulty of an OOD generalization problem.

- We prove novel OOD generalization error bounds based on our formulation. The upper and lower bounds together indicate that the expansion function well characterizes the OOD generalization ability of features with different levels of variation.

- We design a model selection criterion that is inspired by our generalization bounds. Our criterion takes both the performance on training domains and the variation of models into consideration and is predictive of OOD performance according to our bounds. Experimental results demonstrate our selection criterion can choose models with higher OOD accuracy.

The rest of the paper is organized as follows: Section 2 is our preliminary. In Section 3, we give our theoretical formulation. Section 4 gives our generalization bound. We propose our model selection method in Section 5. In Section 6 we conduct experiments on expansion function and model selection. We review more related works in Section 7 and conclude our work in Section 8.

## 2 Preliminary

Throughout the paper, we consider a multi-class classification task $\mathcal{X} \to \mathcal{Y} = \{1, \ldots, K\}$.[1] Let $\mathcal{E}_{all}$ be the domain set we want to generalize to, and $\mathcal{E}_{avail} \subseteq \mathcal{E}_{all}$ be the available domain set, i.e., all domains we have during the training procedure. We denote $(X^e, Y^e)$ to be the input-label pair drawn from the data distribution of domain $e$. The OOD generalization goal is to find a classifier $f^*$ that minimizes the *worst-domain* loss on $\mathcal{E}_{all}$:

$$f^* = \operatorname*{argmin}_{f \in \mathcal{F}} \mathcal{L}(\mathcal{E}_{all}, f), \ \mathcal{L}(\mathcal{E}, f) \triangleq \max_{e \in \mathcal{E}} \mathbb{E}\big[\ell\big(f(X^e), Y^e\big)\big] \tag{1}$$

where $\mathcal{F} : \mathcal{X} \to \mathbb{R}^K$ is the the hypothetical space and $\ell(\cdot, \cdot)$ is a loss function. Similar to previous works [5, 16, 27, 33], we assume that $f$ can be decomposed into $g \circ h$, where $g \in \mathcal{G} : \mathbb{R}^d \to \mathbb{R}^K$ is the top classifier and $h : \mathcal{X} \to \mathbb{R}^d$ is a $d$-dimensional feature extractor, i.e.,

$$h(x) = (\phi_1(x), \phi_2(x), \ldots, \phi_d(x))^\top, \quad \phi_i \in \Phi.$$

Here $\Phi$ is the set of scalar feature maps which map $\mathcal{X}$ to $\mathbb{R}$ and $d$ is fixed. We will call each $\phi \in \Phi$ a feature for simplicity. Given a domain $e$, we denote the $d$-dimensional random vector $h(X^e)$ as $h^e$, one-dimensional feature $\phi(X^e)$ as $\phi^e$, and the conditional distribution of $h^e, \phi^e$ given $Y^e = y$ as $\mathbb{P}(h^e|y), \mathbb{P}(\phi^e|y)$. For simplicity, we assume the data distribution is balanced in every domain, i.e., $P(Y^e = y) = \frac{1}{K}, \forall y \in \mathcal{Y}, e \in \mathcal{E}_{all}$. Our framework can be easily extended to the case where the balanced assumption is removed, with an additional term corresponding to the imbalance adding to the generalization bounds.

## 3 Framework of OOD Generalization Problem

The main challenge of formalizing the OOD generalization problem is to mathematically describe the connection between $\mathcal{E}_{avail}$ and $\mathcal{E}_{all}$ and how generalization depends on this relation. Towards this goal, we introduce several quantities to characterize the relation of *feature distributions* over different domains and bridge $\mathcal{E}_{avail}$ and $\mathcal{E}_{all}$ by expansion function (Definition 3.3) over the quantities we have introduced. Our framework is motivated by the understanding that, in an OOD generalization task, certain "property" of "good" features in $\mathcal{E}_{avail}$ should be "preserved" in $\mathcal{E}_{all}$ (the reason is described in Section 1). In Section 3.1, we will go into details on what we mean by "property" (variation, Definition 3.1), "good" (informativeness, Definition 3.2), and "preserved" (measured by expansion function). In Section 6.2, we further illustrate the key concepts in our framework by a real-world OOD problem.

### 3.1 Formalizing OOD Problem by Quantifying Feature Distribution

We first introduce the concepts "variation" and "informativeness" of a feature $\phi$. The first one is what we expect to be preserved in $\mathcal{E}_{all}$ and the second one characterizes what features will be considered. Specifically, let $\rho(\mathbb{P}, \mathbb{Q})$ be a symmetric "distance" of two distributions. Note that $\rho$ can have many choices, like $L_2$ Distance, Total Variation and symmetric KL-divergence, etc. The variation and informativeness are defined as follows:

**Definition 3.1** (Variation). *The variation of feature $\phi(\cdot)$ across a domain set $\mathcal{E}$ is*

$$\mathcal{V}_\rho(\phi, \mathcal{E}) = \max_{y \in \mathcal{Y}} \sup_{e, e' \in \mathcal{E}} \rho\big(\mathbb{P}(\phi^e|y), \mathbb{P}(\phi^{e'}|y)\big). \tag{2}$$

*A feature $\phi(\cdot)$ is $\varepsilon$-invariant across $\mathcal{E}$, if $\varepsilon \geq \mathcal{V}(\phi, \mathcal{E})$ (We omit the subscript $\rho$ in case of no ambiguity).*

**Definition 3.2** (Informativeness). *The informativeness of feature $\phi(\cdot)$ across a domain set $\mathcal{E}$ is*

$$\mathcal{I}_\rho(\phi, \mathcal{E}) = \frac{1}{K(K-1)} \sum_{\substack{y \neq y' \\ y, y' \in \mathcal{Y}}} \min_{e \in \mathcal{E}} \rho\big(\mathbb{P}(\phi^e|y), \mathbb{P}(\phi^e|y')\big). \tag{3}$$

*A feature $\phi(\cdot)$ is $\delta$-informative across $\mathcal{E}$, if $\delta \leq \mathcal{I}(\phi, \mathcal{E})$.*

---

[1]Note that our framework can be generalized to other kinds of problems easily.

The variation $\mathcal{V}(\phi, \mathcal{E})$ measures the stability of $\phi(\cdot)$ over the domains in $\mathcal{E}$ and the informativeness $\mathcal{I}(\phi, \mathcal{E})$ captures the ability of $\phi(\cdot)$ to distinguish different labels. We would like to highlight that the variation and informativeness are defined on each one-dimensional feature $\phi(\cdot)$. Unlike previous distance between distributions defined in $d$-dimensional space, our definitions are more reasonable and practical, since it can be easily calculated and analyzed.

We are now ready to introduce the core quantity for connecting $\mathcal{E}_{avail}$ and $\mathcal{E}_{all}$. Our motivation, as elaborated in the introduction section, is that, if a feature is informative for the classification task and invariant over $\mathcal{E}_{avail}$, then to enable OOD generalization from $\mathcal{E}_{avail}$ to $\mathcal{E}_{all}$, it should be still invariant over $\mathcal{E}_{all}$. So the relation between $\mathcal{V}(\phi, \mathcal{E}_{avail})$ and $\mathcal{V}(\phi, \mathcal{E}_{all})$ of an informative feature captures the feasibility and difficulty of OOD generalization. To quantitatively measure this relation, we define the following function class:

**Definition 3.3** (Expansion Function). *We say a function* $s : \mathbb{R}^+ \cup \{0\} \to \mathbb{R}^+ \cup \{0, +\infty\}$ *is an expansion function, iff the following properties hold: 1)* $s(\cdot)$ *is monotonically increasing and* $s(x) \geq x, \forall x \geq 0$*; 2)* $\lim_{x \to 0^+} s(x) = s(0) = 0$.

This function class gives a full characterization of how the variation between $\mathcal{E}_{avail}$ and $\mathcal{E}_{all}$ is related. Based on this function class, we can introduce our formulation of the learnability of OOD generalization as follows:

**Definition 3.4** (Learnability). *Let* $\Phi$ *be the feature space and* $\rho$ *be a distribution distance. We say an OOD generalization problem from* $\mathcal{E}_{avail}$ *to* $\mathcal{E}_{all}$ *is* learnable *if there exists an expansion function* $s(\cdot)$ *and* $\delta \geq 0$*, such that: for all* $\phi \in \Phi$ *satisfying* $\mathcal{I}_\rho(\phi, \mathcal{E}_{avail}) \geq \delta$*, we have* $s(\mathcal{V}_\rho(\phi, \mathcal{E}_{avail})) \geq \mathcal{V}_\rho(\phi, \mathcal{E}_{all})$*. If such* $s(\cdot)$ *and* $\delta$ *exist, we further call this problem* $(s(\cdot), \delta)$*-learnable. If an OOD generalization problem is not learnable, we call it* unlearnable.

To understand the intuition and rationality of our formulation, several discussions are in order.

**Properties of the expansion function.** In Definition 3.3, we highlight two properties of the expansion function. The first property comes naturally from the monotonicity properties of variation: any $\varepsilon_1$-invariant feature is also $\varepsilon_2$-invariant for $\varepsilon_2 \geq \varepsilon_1$; and $\mathcal{V}(\phi, \mathcal{E}_1) \leq \mathcal{V}(\phi, \mathcal{E}_2)$ for any $\mathcal{E}_1 \subseteq \mathcal{E}_2$. The monotonicity also implies that larger $\mathcal{E}_{all}$ will induce larger $s(\cdot)^2$ and it is also harder to be generalized to. From this view, we can see that the scale of $s(\cdot)$ can reflect the difficulty of OOD generalization. The second property is more crucial since it formulates the intuition that if an informative feature is *almost* invariant in $\mathcal{E}_{avail}$, it should remain invariant in $\mathcal{E}_{all}$. Without this assumption, OOD generalization can never be guaranteed because we cannot predict whether an invariant and informative feature in $\mathcal{E}_{avail}$ will vary severely in *unseen* $\mathcal{E}_{all}$.

**Necessity of informativeness.** We include a seemingly redundant quantity informativeness in the definition of learnability. However, this term is necessary because only informative features are responsible for the performance of classification. Non-informative but invariant features over $\mathcal{E}_{avail}$ may only capture some noise that is irrelevant to the classification problem, and we shall not expect the noise to be invariant over $\mathcal{E}_{all}$. Moreover, we show in Figure 1 that in practice, many invariant but useless features in $\mathcal{E}_{avail}$ vary a lot in $\mathcal{E}_{all}$, and adding the constraint of informativeness makes the expansion function reasonable. In addition, there are multiple choices of $(s(\cdot), \delta)$ to make an OOD generalization problem learnable: larger $\delta$ will filter out more features, and so $s(\cdot)$ can be smaller (flatter). This multiplicity will result in a trade-off between $s(\cdot)$ and $\delta$, which will be discussed in Section 6.2.

**Two extreme cases: *i.i.d.* & unlearnable.** To better understand the concept of learnability, we consider two extreme cases. (1) The first example is when all data from different $e \in \mathcal{E}_{all}$ are *identically* distributed, i.e., the classic supervised learning setting. This problem is $(s(\cdot), 0)$-learnable with $s(x) = x$, implying no extra difficulty in OOD generalization. (2) As an example of unlearnable, consider the following case (modified from Colored MNIST [5]): For $e \in \mathcal{E}_{avail}$, images with label 0 always has a red background while images with label 1 has a blue background. For $e \in \mathcal{E}_{all} \setminus \mathcal{E}_{avail}$, this relationship is entirely inverse. Since data from different $e \in \mathcal{E}_{avail}$ are identically distributed but different from other $e \in \mathcal{E}_{all}$, no expansion function can make it learnable, i.e., it is OOD-unlearnable.

---

[2]When we talk about the scale of $s(\cdot)$, e.g. it is larger / smaller, we mean the comparison of two expansion function, rather than the comparison along a function.

The unlearnability of this case also coincides with our intuition: Without prior knowledge, it is not clear from merely the training data, whether the task is to distinguish digit 0 from 1, or to distinguish color red from blue. As a result, generalization to $\mathcal{E}_{all}$ cannot be guaranteed.

## 4 Generalization Bound

In this section, we consider an OOD generalization problem from $\mathcal{E}_{avail}$ to $\mathcal{E}_{all}$, and our goal is to analyze the OOD generalization error of classifier $f = g \circ h$ defined by

$$\text{err}(f) = \mathcal{L}(\mathcal{E}_{all}, f) - \mathcal{L}(\mathcal{E}_{avail}, f),$$

where we assume the loss function $l(\cdot, \cdot)$ is bounded by $[0, C]$. We prove two upper bounds (4.1, 4.2) as well as a lower bound (4.3) for $\text{err}(f)$ based on our formulation. Our bounds together provide a complete characterization of the difficulty of OOD generalization. Since we expect that an invariant classifier can generalize to unseen domains, we hope to bound $\text{err}(f)$ in terms of the certain variation of $f$. To this end, we define the variation and informativeness of $f$ in terms of its features, i.e.,

$$\mathcal{V}^{\text{sup}}(h, \mathcal{E}_{avail}) \triangleq \sup_{\beta \in \mathcal{S}^{d-1}} \mathcal{V}(\beta^\top h, \mathcal{E}_{avail}),$$

$$\mathcal{I}^{\text{inf}}(h, \mathcal{E}_{avail}) \triangleq \inf_{\beta \in \mathcal{S}^{d-1}} \mathcal{I}(\beta^\top h, \mathcal{E}_{avail}),$$

where $(\beta^\top h)(x) = \beta^\top h(x)$ is a feature and $\mathcal{S}^{d-1} = \{\beta \in \mathbb{R}^d : \|\beta\|_2 = 1\}$ is the unit $(d-1)$-sphere.

**Necessity of using supremum over linear combination.** One seemingly plausible definition of the variation of a classifier $f$ can be the supremum over all $\mathcal{V}(\phi_i, \mathcal{E}_{avail}), i \in [d]$. However, as is shown in Appendix 1, it is possible that two high-dimensional joint distributions have close marginal distribution in each dimension, while they do not overlap. In other words, there exist cases where $\mathcal{V}(\phi_i, \mathcal{E}_{all}) = 0, \forall i \in [d]$ but after applying the top model $g$ over $\phi_i$'s, the distribution varies a lot in $\mathcal{E}_{avail}$. Our definition comes from the simple idea that the class of top model $\mathcal{G}$ is at least a linear space, so we should at least consider the variation of every (normalized) linear combination of $h(\cdot)$. With this, we can guarantee the joint distribution distance is still small.

**Theorem 4.1** (Main Theorem). *Suppose we have learned a classifier $f(x) = g(h(x))$ such that $\forall e \in \mathcal{E}_{all}$ and $\forall y \in \mathcal{Y}, p_{h^e|Y^e}(h|y) \in L^2(\mathbb{R}^d)$. Denote the characteristic function of random variable $h^e|Y^e$ as $\hat{p}_{h^e|Y^e}(t|y) = \mathbb{E}[\exp\{i\langle t, h^e\rangle\}|Y^e = y]$. Assume the hypothetical space $\mathcal{F}$ satisfies the following regularity conditions that $\exists \alpha, M_1, M_2 > 0, \forall f \in \mathcal{F}, \forall e \in \mathcal{E}_{all}, y \in \mathcal{Y}$,*

$$\int_{h \in \mathbb{R}^d} p_{h^e|Y^e}(h|y)|h|^\alpha \mathrm{d}h \leq M_1 \quad \text{and} \quad \int_{t \in \mathbb{R}^d} |\hat{p}_{h^e|Y^e}(t|y)||t|^\alpha \mathrm{d}t \leq M_2. \tag{4}$$

*If $(\mathcal{E}_{avail}, \mathcal{E}_{all})$ is $\big(s(\cdot), \mathcal{I}^{inf}(h, \mathcal{E}_{avail})\big)$-learnable under $\Phi$ with Total Variation $\rho$[3], then we have*

$$\text{err}(f) \leq O\Big(s\big(\mathcal{V}_\rho^{sup}(h, \mathcal{E}_{avail})\big)^{\frac{\alpha^2}{(\alpha+d)^2}}\Big). \tag{5}$$

*Here $\rho$ is total variation distance, and $O(\cdot)$ depends on $d, C, \alpha, M_1, M_2$.*

The above theorem holds for a general classifier learned by any algorithms. Due to its generality, we need to introduce some technical regularity conditions on the density function. The assumption (4) assume the decay rate of density and its characteristic function, which is common in the literature, e.g. [14]. This theorem demonstrates that, the generalization error can be bounded by a function of the variation of $h$, and it converges to 0 as the variation approaches to 0. Under some special but typical case where the top model $g$ is linear, we can further show that even without the regularity conditions in Theorem 4.1, we have a much better (linear) convergence rate.

**Theorem 4.2** (Linear Top Model). *Consider any loss satisfying $\ell(\hat{y}, y) = \sum_{k=1}^K \ell_0(\hat{y}_k, y_k)$.[4] For any classifier with linear top model $g$, i.e.,*

$$f(x) = Ah(x) + b \quad \text{with} \quad A \in \mathbb{R}^{K \times d}, \ b \in \mathbb{R}^K,$$

---

[3]For two distribution $\mathbb{P}, \mathbb{Q}$ with probability density function $p, q$, $\rho(\mathbb{P}, \mathbb{Q}) = \frac{1}{2}\int_x |p(x) - q(x)|\mathrm{d}x$.

[4]This decomposition is a technical assumption to make the proof more convenient. Truncated square loss or Truncated absolute loss satisfy this assumption.

if $(\mathcal{E}_{avail}, \mathcal{E}_{all})$ is $\big(s(\cdot), \mathcal{I}^{inf}(h, \mathcal{E}_{avail})\big)$-learnable under $\Phi$ with Total Variation $\rho$, then we have

$$\mathrm{err}(f) \leq O\Big(s\big(\mathcal{V}^{sup}(h, \mathcal{E}_{avail})\big)\Big). \tag{6}$$

*Here $O(\cdot)$ depends only on $d$ and $C$.*

**Discussion.** Theorem 4.1 shows that, for any model, the generalization gap depends largely on the model's variation captured by $\mathcal{V}^{\mathrm{sup}}(h, \mathcal{E}_{avail})$. The result is irrelevant to the algorithm and provides a guarantee for the generalization gap from $\mathcal{E}_{avail}$ to $\mathcal{E}_{all}$, so long as the learned model $f$ is invariant, i.e. $\mathcal{V}^{\mathrm{sup}}(h, \mathcal{E}_{avail})$ is small. When $s(\cdot)$ is fixed, a model with smaller $\mathcal{V}^{\mathrm{sup}}(h, \mathcal{E}_{avail})$ results in a smaller gap, which matches our understanding that invariant features in $\mathcal{E}_{avail}$ are somehow invariant in $\mathcal{E}_{all}$. When $\mathcal{V}^{\mathrm{sup}}(h, \mathcal{E}_{avail})$ is fixed, more difficult generalization will generate a larger expansion function, which leads to a larger gap. For the Gaussian class with bounded mean and variance, $\alpha \gg d$ and the convergent rate is almost linear.

However, without any constraint to $g$, the convergent rate might be small. Theorem 4.2 then offers a generalization bound with a *linear* convergent rate under mild assumptions when $g$ is linear, which is common in reality. It relaxes the concentration assumption (Formula 4) and asks only for the integrability of the density. The convergent rate is identical to the convergent rate of the expansion function, showing that $s(\cdot)$ captures the generalization quite well.

**Proof Sketch of Theorem 4.1.** The proof of the main result, Theorem 4.1, is decomposed into the following steps. First, we transform $\mathrm{err}(f)$ into the Total Variation of joint distributions of features in different domains (Step 1). To bound the Total Variation, it is sufficient to bound the distance of the corresponding Fourier transform, and further, it is equivalent to bound the Radon transform of joint distributions (Step 2). Eventually, we show that $\mathcal{V}^{\mathrm{sup}}(\beta^{\top}h, \mathcal{E}_{avail})$ can be used to bound the Radon transform, which finishes the proof (Step 3).

**Step 1**. The OOD generalization error can be bounded as:

$$\mathrm{err}(f) \leq \sup_{(e,e')\in(\mathcal{E}_{avail}, \mathcal{E}_{all})} \frac{C}{K} \sum_{y\in\mathcal{Y}} \int_{h\in\mathbb{R}^d} \big|p_{h^e|Y^e}(h|y) - p_{h^{e'}|Y^{e'}}(h|y))\big| \mathrm{d}h. \tag{7}$$

**Step 2**. According to the assumption (4), the dominant term in (7) is

$$\int_{|h|\leq r_1} \Big| \int_{|t|\leq r_2} e^{-i\langle h, t\rangle} \big(\hat{p}_{h^e|Y^e}(t|y) - \hat{p}_{h^{e'}|Y^{e'}}(t|y)\big)\big) \mathrm{d}t \Big| \mathrm{d}t, \tag{8}$$

where $r_1$ and $r_2$ are well-selected scalars that depend on $s\big(\mathcal{V}_{\rho}^{\mathrm{sup}}(h, \mathcal{E}_{avail})\big)$. By the Projection Theorem [31, 42] and the Fourier Inversion Formula, (8) is bounded above by

$$O(r_1^d r_2^d) \times \int_{u\in\mathbb{R}} \big|\mathcal{R}_{e'}(\beta, u) - \mathcal{R}_e(\beta, u)\big| \mathrm{d}u, \tag{9}$$

where $\mathcal{R}_e(\beta, u)$ is the Radon transform of $p_{h^e|Y^e}(t|y)$.

**Step 3**. The right-hand side of Formula 8 can be bounded by $O\big(r_1^d r_2^d s\big(\mathcal{V}_{\rho}^{\mathrm{sup}}(h, \mathcal{E}_{avail})\big)\big)$. We finish the proof by selecting appropriate $r_1$ and $r_2$ to balance the rate of the dominant term and other minor terms. For more details, please see Appendix 2 for the complete proofs.

Now we turn to the lower bound of $\mathrm{err}(f)$.

**Theorem 4.3** (Lower Bound). *Consider 0-1 loss: $\ell(\hat{y}, y) = \mathbb{I}(\hat{y} \neq y)$. For any $\delta > 0$ and any expansion function satisfying 1) $s'_+(0) \triangleq \lim_{x\to 0^+} \frac{s(x)-s(0)}{x} \in (1, +\infty)$; 2) exists $k > 1, t > 0$, s.t. $kx \leq s(x) < +\infty, x \in [0, t]$, there exists a constant $C_0$ and an OOD generalization problem $(\mathcal{E}_{avail}, \mathcal{E}_{all})$ that is $(s(\cdot), \delta)$-learnable under linear feature space $\Phi$ w.r.t symmetric KL-divergence $\rho$, s.t. $\forall \varepsilon \in [0, \frac{t}{2}]$, the optimal classifier $f$ satisfying $\mathcal{V}^{sup}(h, \mathcal{E}_{avail}) = \varepsilon$ will have the OOD generalization error lower bounded by*

$$\mathrm{err}(f) \geq C_0 \cdot s(\mathcal{V}^{sup}(h, \mathcal{E}_{avail})). \tag{10}$$

Theorem 4.3 shows that $\mathrm{err}(f)$ of optimal classifier $f$ is lower bounded by its variation. Here "optimal" means the classifier that minimize $\mathcal{L}(f, \mathcal{E}_{avail})$. Altogether, the above three theorems

offer a bidirectional control of OOD generalization error, showing that our formulation can offer a fine-grained description of most OOD generalization problem in a theoretical way. To pursue a good OOD performance, OOD algorithm should focus on improving predictive performance on $\mathcal{E}_{avail}$ and controlling the variation $\mathcal{V}^{\text{sup}}(h, \mathcal{E}_{avail})$ simultaneously. Note that this bound starts from population error, and we call for future works to combine our generalization bound and traditional bound from data samples to population error, giving a more complete characterization of the problem.

## 5  Variation as a Factor of Model Selection Criterion

As is pointed out in [21], model selection has a significant effect on domain generalization, and any OOD algorithm without a model selection criterion is not complete. [21] trained more than 45,900 models with different algorithms, and results show that when traditional selection methods are applied, none of OOD algorithms can outperform ERM [58] by a significant margin. This result is not strange, since traditional selection methods focus mainly on (validation) accuracy, which is biased in OOD generalization [21, 63]. A very typical example is Colored MNIST [5], where the image is colored according to the label, but the relationship varies across domains. As explained in [5], ERM principle will only capture this spurious feature (color) and performs badly in $\mathcal{E}_{all}$. Since ERM is exactly minimizing loss in $\mathcal{E}_{avail}$, any model selection method using validation accuracy alone is likely to choose ERM rather than any other OOD algorithm [63]. Thus no algorithm will have a significant improvement compared to ERM.

A natural question arises: what else can we use, in addition to accuracy? Theorem 4.1 points out that, learning feature with small variation across $\mathcal{E}_{avail}$ is important for decreasing OOD generalization error. Once a model $f$ achieves a small $\mathcal{V}^{\text{sup}}(h, \mathcal{E}_{avail})$, then $\text{err}(f)$ will be small. If the validation accuracy is also high, we shall know that the OOD accuracy will remain high. To this end, we propose our heuristic selection criterion (Algorithm 1). Instead of considering validation accuracy alone, we combine it with feature variation and *select the model with high validation accuracy as well as low variation.*

---

**Algorithm 1:** Model Selection

**Input:** available dataset $\mathcal{X}_{avail} = (\mathcal{X}_{train}, \mathcal{X}_{val})$, candidate models set $\mathcal{M}$, var_acc_rate $r_0$.
**for** $f = g \circ h$ *in* $\mathcal{M}$ **do**
    **for** $i$ *in* $[d]$ **do**
        $\hat{\mathcal{V}}_i \leftarrow \max_{y \in \mathcal{Y}, \mathcal{X}^e \neq \mathcal{X}^{e'} \in \mathcal{X}_{avail}} \text{Total Variation}(\mathbb{P}(\phi_i^e | y), \mathbb{P}(\phi_i^{e'} | y));$     ▷Use GPU KDE
    **end**
    $\mathcal{V}_f \leftarrow \text{mean}_{i \in [d]} \hat{\mathcal{V}}_i$
    $\text{Acc}_f \leftarrow$ compute validation accuracy of $f$ using $\mathcal{X}_{val}$
**end**
**Return** $\text{argmax}_{f \in \mathcal{M}}(\text{Acc}_f - r_0 \mathcal{V}_f)$

---

We briefly explain Algorithm 1 here. For each candidate model, we calculate its variation using the average of each feature's variation, i.e., $\frac{1}{d} \sum_{i \in [d]} \mathcal{V}(\phi_i, \mathcal{X}_{avail})$. When deriving the bounds, we use $\mathcal{V}^{\text{sup}}$ instead of their average because we need to consider *the worst case*, i.e., the worst top model. In practice, we find out that the average of $\mathcal{V}(\phi_i, \mathcal{X}_{avail})$ is enough to improve selection.

Our criterion of model selection is

$$\text{Acc}_f - r_0 \mathcal{V}_f, \tag{11}$$

i.e., we select a model with high validation accuracy and low variation *simultaneously*. Here $r_0$ is a hyper-parameter representing the concrete relationship between $\text{err}(f)$ and $\mathcal{V}_f$. Although we have already used one hyper-parameter to help select multiple hyper-parameter combinations, it is natural to ask whether we can further get rid of the selection of $r_0$. Since $r_0$ represents the relationship between variation and accuracy, which is actually determined by the unknown expansion function, explicitly calculating $r_0$ is not possible. However, we can empirically estimate $r_0$ using $r_0 = \frac{\text{Std}_{f \in \hat{\mathcal{M}}} \text{Acc}_f}{\text{Std}_{f \in \hat{\mathcal{M}}} \mathcal{V}_f}$, where $\hat{\mathcal{M}} \subset \mathcal{M}$ is the model with not bad validation accuracy. We do not use the whole set $\mathcal{M}$ because some OOD algorithms will perform extremely bad when the penalty is

huge, and these models will influence our estimation of the ratio. Since high validation means large informativeness in learned features, the use of $\hat{\mathcal{M}}$ is an implicit application of informative assumption.

As shown in Section 6.1, our method can select models with higher OOD accuracy in various OOD datasets. We also explain in Appendix 3 why our method can outperform the traditional method in Color MNIST, where the dataset is hand-make and simple enough to calculate the expansion function.

# 6 Experiments

In this section, we conduct experiments to compare our model selection criterion (Section 5) with the baseline method[5] [21]. Since both the variation and informativeness in Definition 3.1 are based on one-dimensional features, we can directly estimate these quantities feature-by-feature and design model selection method based on them. To verify the existence of the expansion function and to see what it's like in a real-world dataset, we plot nearly 2 million features trained in a common-used OOD dataset and compute their variation and informativeness. We then draw the expansion function for this problem.

## 6.1 Experiments on Model Selection

In this section, we conduct experiments to compare the performance of models selected by our method and by validation accuracy. We train models on different datasets, different $\mathcal{E}_{avail}$, and select models according to a different selection criteria. We then compare the OOD accuracy of selected models.

**Settings** We train our model on three benchmark OOD datasets (PACS [34], OfficeHome [59], VLCS [57]) and consider all possible selections of $(\mathcal{E}_{avail}, \mathcal{E}_{all})$ . We choose ResNet–50 as our network architecture. We use ERM [58] and four common-used OOD algorithms (CORAL [55], Inter-domain Mixup [62], Group DRO [51], and IRM [5]). For each environment setup, we train 200 models using different algorithms, penalties, learning rates, and epoch. After training, we employ different selection methods and compare the OOD accuracy of the selected models. As stated in Section 5, we use the standard deviation of $\mathcal{V}$ and validation accuracy in $\hat{\mathcal{M}}$ to estimate $r_0$, where $\hat{\mathcal{M}} = \{f \in \mathcal{M} : \text{Acc}_f \geq \max_{\hat{f}} \text{Acc}_{\hat{f}} - 0.1\}$. Note that calculating $\mathcal{V}(\phi_i, \mathcal{X}_{avail})$ takes calculus many times, so we design a parallel GPU kernel density estimation to speed up the whole process a hundred times and manage to finish one model in seconds. For more details about the experiments, see Appendix 4.

Table 1: Model Selection Result. "Env" denotes the unseen domain during training. "Val" denotes the OOD accuracy of model selected by validation accuracy.

| | Env | A | C | P | S | avg | acc inc |
|---|---|---|---|---|---|---|---|
| PACS | Val | 85.20% | 80.42% | 96.17% | 77.86% | 84.91% | - |
| | Ours | **88.72%** | **81.74%** | **96.83%** | **79.00%** | **86.57%** | 1.66%↑ |
| | Env | A | C | P | R | avg | acc inc |
| OfficeHome | Val | 61.85% | **55.56%** | 74.72% | 76.25% | 67.09% | - |
| | Ours | **65.76%** | 55.07% | **75.20%** | **76.31%** | **68.09%** | 1.00%↑ |
| | Env | C | L | S | V | avg | acc inc |
| VLCS | Val | 97.46% | 64.83% | **69.50%**[6] | **70.97%** | 75.69% | - |
| | Ours | **97.81%** | **66.98%** | **69.50%** | **70.97%** | **76.32%** | 0.63%↑ |

**Result** We summarize our experimental results in Table 1. For each environment setup, we select the best model according to Algorithm 1 and validation accuracy. The results show that on all datasets, our selection criterion significantly outperforms the validation accuracy in average OOD accuracy. For a more detailed comparison, our method improves the OOD accuracy in most of the 12 setups. Our experiments demonstrate that our criterion can help select models with higher OOD accuracy.

---

[5]Our experiments is conducted in DomainBed: `https://github.com/facebookresearch/DomainBed`.
[6]Notice that some OOD accuracy are the same in the two methods since the same model is selected. This happens when the unseen domain is close to $\mathcal{E}_{avail}$ so that the validation accuracy metric is close to ours.

## 6.2 Learnability of Real-World OOD Problem

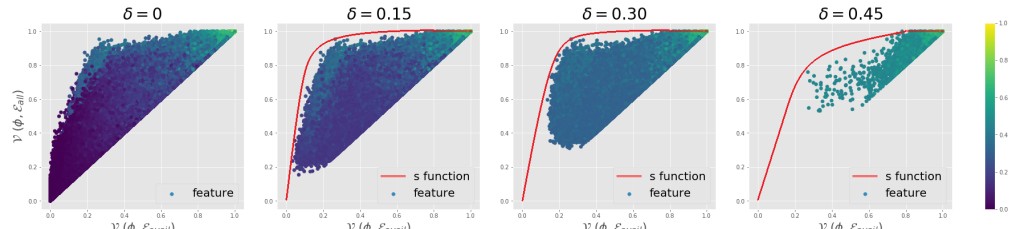

Figure 1: The expansion function of the OOD generalization problem on Office-Home. The x-axis stands for $\mathcal{V}(\phi, \mathcal{E}_{avail})$ and the y-axis for $\mathcal{V}(\phi, \mathcal{E}_{all})$. There are approximately 2 million points in each image, with each point representing a feature, and its color represents its informativeness. The solid red line stands for the expansion function under the corresponding $\delta$. When $\delta$ increases, the expansion function decreases. When $\delta = 0$, no expansion function can make it learnable.

One may wonder if the expansion function really exists and what it will look like for a real-world OOD generalization task. In this section, we consider the OOD dataset Office-Home [59]. We explicitly plot *millions of* features' $\mathcal{V}_\rho(\phi, \mathcal{E}_{avail})$ and $\mathcal{V}_\rho(\phi, \mathcal{E}_{all})$ with Total Variation $\rho$ to see what the expansion function is like in this task. We take the architecture as ResNet-50 [23], and we trained thousands of models with more than five algorithms, obtaining about 2 million features. The results are in Figure 1.

**Existence of** $s(\cdot)$. When $\delta = 0$, some non-informative features are nearly 0-invariant across $\mathcal{E}_{avail}$ but are varying across $\mathcal{E}_{all}$, so no expansion function can make this task learnable, i.e., this task is NOT $(s(\cdot), 0)$ for any expansion function. But as $\delta$ increases, only informative features are left, and now we can find appropriate $s(\cdot)$ to make it learnable. We can clearly realize from the figure that $s(\cdot)$ do exist when $\delta \geq 0.15$.

**Trade-off between** $s(\cdot)$ **and** $\delta$. The second phenomenon is that the slope of $s(\cdot)$ decreases as $\delta$ increases, showing a trade-off between $s(\cdot)$ and $\delta$. Although this trade-off comes naturally from the definition of learnability, it has a deep meaning. As is shown in Section 4, err$(f)$ is bounded by $O(s(\varepsilon))$ where $\varepsilon$ is the variation of the model. To make the bound tighter, a natural idea is to choose a flatter $s(\cdot)$. However, a flatter $s(\cdot)$ corresponds to a larger $\delta$. Typically, learning a model to meet this higher informativeness requirement is more difficult, and it is possible that the algorithm achieves this by capturing more domain-specific features, which will therefore increase the variation of the model, $\varepsilon$. As a result, we are not sure whether $s(\varepsilon)$ will increase or decrease. We believe this is also the essence of model selection: i.e., *to trade-off between the variation and informativeness of a model*, which is done in Formula 11.

## 7 More Related Works

Domain generalization [12, 39], or OOD generalization, has drawn much attention recently [21, 30]. The goal is to learn a model from several training domains and expect good performance on unseen test domains. [60, 64] offer a comprehensive survey. A popular solution is to extract domain-invariant feature representation. [45] and [49] proved that when the model is linear, the invariance under training domains can help discover invariant features on test domains. [5] introduces the invariant prediction into neural networks and proposes a practical objective function. After that, a lot of works arise from the view of causal discovery, distributional robustness and conditional independence [1, 7, 16, 15, 26, 32, 33, 43, 51, 61]. On the other hand, some works point out the weakness of existing methods from the theoretical and experimental perspectives [2, 21, 29, 41, 50].

The OOD generalization requires restrictions on how the target domains may differ. A straightforward approach is to define a set of test domains around the training domain using some distribution distance measure [6, 13, 19, 25, 51, 53, 54, 61]. Another feasible route is the causal framework which is robust to the test distributions caused by interventions[44, 46] on variables, e.g., [5, 24, 36, 37, 40, 47, 49, 52]. The principle of these methods is that a causal model is invariant and can achieve the minimal worst-case risk [4, 22, 44, 49]. Since the test distribution is unknown, additional assumptions are required for

generalization analysis. [12, 18, 39] assume that the domains are generated from a hyper-distribution and measures the average risk estimation error bound. [3] derives a risk bound for any linear combination of training domains. For more related results in domain adaptation, a closed field where the test domains can be seen but are unlabeled, please see [9, 10, 28].

# 8 Conclusion

In this paper, we take the first step towards a rigorous theoretical framework of OOD generalization. We propose a mathematical formulation to characterize the learnability of OOD generalization problem. Based on our framework, we prove generalization bounds and give guarantees for OOD generalization error. Inspired by our bound, we design a model selection criterion to check the model's variation and validation accuracy simultaneously. Experiments show that our metric has a significant advantage over the traditional selection method.

## Acknowledgments and Disclosure of Funding

Authors are thankful to the anonymous reviewers for their helpful and constructive feedback.

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
