# Supplementary Materials for "Towards a Theoretical Framework of Out-of-Distribution Generalization"

**Haotian Ye**
Peking University
Pazhou Lab
haotianye@pku.edu.cn

**Chuanlong Xie**
Huawei Noah's Ark Lab
xie.chuanlong@huawei.com

**Tianle Cai**
Peking University
caitianle1998@pku.edu.cn

**Ruichen Li**
Peking University
xk-lrc@pku.edu.cn

**Zhenguo Li**
Huawei Noah's Ark Lab
Li.Zhenguo@huawei.com

**Liwei Wang**
Key Laboratory of Machine Perception,
MOE, School of EECS,
Institute for Artificial Intelligence,
Peking University
wanglw@cis.pku.edu.cn

## 1  Illustration of Model's Variation

In this section, we illustrate why we need to define the variation of a model $f$ as

$$\mathcal{V}^{\sup}(h, \mathcal{E}_{avail}) \quad \triangleq \quad \sup_{\beta \in \mathcal{S}^{d-1}} \mathcal{V}(\beta^\top h, \mathcal{E}_{avail}),$$

where $(\beta^\top h)(x) = \beta^\top h(x)$ and $\mathcal{S}^{d-1} = \{\beta \in \mathbb{R}^d : \|\beta\|_2 = 1\}$ is the unit $(d-1)$-sphere.

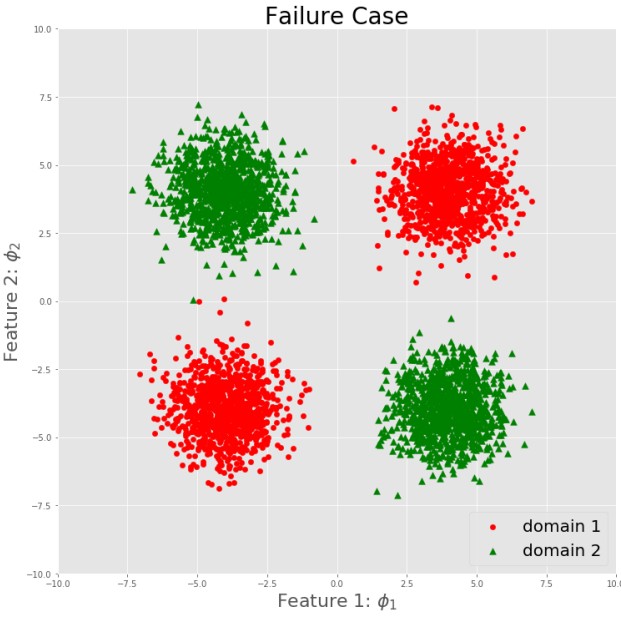

Figure 1: The Failure Case.

35th Conference on Neural Information Processing Systems (NeurIPS 2021).

One seemingly plausible definition of the variation of a classifier $f$ can be the supremum over all $\mathcal{V}(\phi_i, \mathcal{E}_{avail}), i \in [d]$. However, there exist cases where $\mathcal{V}(\phi_i, \mathcal{E}_{all}) = 0, \forall i \in [d]$ but the distribution of $h$ varies a lot in $\mathcal{E}_{avail}$. We give a concrete failure case here.

Consider a binary classification task with $\mathcal{Y} = \{-1, 1\}$ and let $d = 2$. Assume we learn a feature extractor $h = (\phi_1, \phi_2)^\top$ such that for a given label $y$, the distributions of $h$ under two domains are

$$\text{Domain 1:} \quad y \sim \text{unif}\{+1, -1\}, \quad h|y \sim \mathcal{N}\big(y(4, 4)^\top, \mathbf{I}_2\big)$$
$$\text{Domain 2:} \quad y \sim \text{unif}\{+1, -1\}, \quad h|y \sim \mathcal{N}\big(y(4, -4)^\top, \mathbf{I}_2\big).$$

It is easy to see that the marginal distributions of both features alone are identical across the two domains. However, the distributions of $h$ are different (nearly separate at all). The empirical distributions of the two domains are present in Figure 1. This example shows that merely control the supremum of $\mathcal{V}(\phi_i, \mathcal{E}_{avail}), i \in [d]$ is not enough to control the Total Variation of two domains' density, and so it is not enough to upper bound the $\text{err}(f)$. To do so, we need a stronger quantity like $\mathcal{V}^{\sup}(h, \mathcal{E}_{avail})$.

## 2 Proofs

In this section, we provide complete proofs of our three bounds.

### 2.1 Proof of Theorem 4.1

**Theorem** *Let the loss function $\ell(\cdot, \cdot)$ be bounded by $[0, C]$. We denote*

$$\mathcal{V}^{sup}(h, \mathcal{E}_{avail}) \triangleq \sup_{\beta \in \mathcal{S}^{d-1}} \mathcal{V}(\beta^\top h, \mathcal{E}_{avail}),$$

$$\mathcal{I}^{inf}(h, \mathcal{E}_{avail}) \triangleq \inf_{\beta \in \mathcal{S}^{d-1}} \mathcal{I}(\beta^\top h, \mathcal{E}_{avail}),$$

*where $(\beta^\top h)(x) = \beta^\top h(x)$ is a feature and $\mathcal{S}^{d-1} = \{\beta \in \mathbb{R}^d : \|\beta\|_2 = 1\}$ is the unit $(d-1)$-sphere. Suppose we have learned a classifier $f(x) = g(h(x))$ such that for any $e \in \mathcal{E}_{all}$ and $y \in \mathcal{Y}$, $p_{h^e|Y^e}(h|y) \in L^2(\mathbb{R}^d)$. Denote the characteristic function of random variable $h^e|Y^e$ as*

$$\hat{p}_{h^e|Y^e}(t|y) = \mathbb{E}[\exp\{i\langle t, h^e\rangle\}|Y^e = y].$$

*Assume the hypothetical space $\mathcal{F}$ satisfies the following regularity conditions that $\exists \alpha, M_1, M_2 > 0, \forall f \in \mathcal{F}, \forall e \in \mathcal{E}_{all}, y \in \mathcal{Y}$,*

$$\int_{h \in \mathbb{R}^d} p_{h^e|Y^e}(h|y)|h|^\alpha \mathrm{d}h \leq M_1 \quad and \quad \int_{t \in \mathbb{R}^d} \hat{p}_{h^e|Y^e}(t|y)|t|^\alpha \mathrm{d}t \leq M_2. \tag{1}$$

*If $(\mathcal{E}_{avail}, \mathcal{E}_{all})$ is $\big(s(\cdot), \mathcal{I}^{inf}(h, \mathcal{E}_{avail})\big)$-learnable under $\Phi$ with Total Variation $\rho$[1], then we have*

$$\text{err}(f) \leq O\Big(s\big(\mathcal{V}_\rho^{sup}(h, \mathcal{E}_{avail})\big)^{\frac{\alpha^2}{(\alpha+d)^2}}\Big).$$

*Here $O(\cdot)$ depends on $d, C, \alpha, M_1, M_2$.*

*Proof.* For any $e \in \mathcal{E}_{avail}$ and $e' \in \mathcal{E}_{all}$,

$$\mathbb{P}_Y(y) = \mathbb{P}_{Y^e}(y) = \mathbb{P}_{Y^{e'}}(y).$$

We can decompose the loss gap between $e$ and $e'$ as

$$\mathbb{E}\big[\ell(f(X^{e'}), Y^{e'})\big] - \mathbb{E}\big[\ell(f(X^e), Y^e)\big]$$
$$= \mathbb{E}\big[\ell(g(h(X^{e'})), Y^{e'})\big] - \mathbb{E}\big[\ell(g(h(X^e)), Y^e)\big]$$
$$= \sum_{y=1}^K \mathbb{P}_Y(y)\Big(\mathbb{E}\big[\ell(g(h(X^{e'})), Y^{e'})\big|Y^{e'} = y\big] - \mathbb{E}\big[\ell(g(h(X^e)), Y^e)\big|Y^e = y\big]\Big).$$

---

[1]For two distribution $\mathbb{P}, \mathbb{Q}$ with probability density function $p, q, \rho(\mathbb{P}, \mathbb{Q}) = \frac{1}{2}\int_x |p(x) - q(x)|\mathrm{d}x$.

Therefore, to bound $\mathrm{err}(f)$, it is sufficient to bound

$$\left|\mathbb{E}\big[\ell(f(X^{e'}), Y^{e'})\big|Y^{e'} = y\big] - \mathbb{E}\big[\ell(f(X^e), Y^e)\big|Y^e = y\big]\right|$$

for any $y \in \mathcal{Y}, (e, e') \in (\mathcal{E}_{avail}, \mathcal{E}_{all})$. Given $y, e, e'$, we have

$$\left|\mathbb{E}\big[\ell(f(X^{e'}), Y^{e'})\big|Y^{e'} = y\big] - \mathbb{E}\big[\ell(f(X^e), Y^e)\big|Y^e = y\big]\right|$$

$$\leq \quad C \int_{\mathbb{R}^d} \big|p_{h^{e'}|Y^{e'}}(h|y) - p_{h^e|Y^e}(h|y)\big|\mathrm{d}h = C * I$$

where $h^e$ represents the $d$-dimensional random vector $h(X^e)$ and

$$I = \int_{\mathbb{R}^d} \big|p_{h^{e'}|Y^{e'}}(h|y) - p_{h^e|Y^e}(h|y)\big|\mathrm{d}h.$$

In the following, we shall show that the term $I$ is upper bounded by $O\Big(s\big(\mathcal{V}^{\sup}(h, \mathcal{E}_{avail})\big)\Big)$.

First, we decomposed the term $I$ into $I_1 + I_2$ where

$$I_1 \quad = \quad \int_{|h| \leq r_1} \big|p_{h^{e'}|Y^{e'}}(h|y) - p_{h^e|Y^e}(h|y)\big|\mathrm{d}h$$

$$I_2 \quad = \quad \int_{|h| > r_1} \big|p_{h^{e'}|Y^{e'}}(h|y) - p_{h^e|Y^e}(h|y)\big|\mathrm{d}h.$$

Here $r_1$ is a scalar to be decided, and $|h|$ is the Euclidean norm of $h$. According to (1), the term $I_2$ is bounded above:

$$I_2 \quad \leq \quad \int_{|h| > r_1} \big|p_{h^{e'}|Y^{e'}}(h|y) - p_{h^e|Y^e}(h|y)\big||h|^\alpha r_1^{-\alpha}\mathrm{d}h$$

$$\leq \quad r_1^{-\alpha}\Big(\int_{h \in \mathbb{R}^d} \big|p_{h^{e'}|Y^{e'}}(h|y)\big||h|^\alpha\mathrm{d}h + \int_{h \in \mathbb{R}^d} \big|p_{h^e|Y^e}(h|y)\big||h|^\alpha\mathrm{d}h\Big)$$

$$\leq \quad 2M_1 r_1^{-\alpha}.$$

Next we deal with $I_1$. Since $p_{h^{e'}|Y^{e'}} \in L^1(\mathbb{R}^d)$ and $\hat{p}_{h^e|Y^e} \in L^1(\mathbb{R}^d)$,

$$p_{h^e|Y^e}(h|y) = \int_{t \in \mathbb{R}^d} e^{-i\langle t, h\rangle}\hat{p}_{h^e|Y^e}(t|y)\mathrm{d}t.$$

Then we have

$$\big|p_{h^{e'}|Y^{e'}}(h|y) - p_{h^e|Y^e}(h|y)\big|$$

$$\leq \quad \Big|\int_{t \in \mathbb{R}^d} \exp(-i\langle t, h\rangle)\big(\hat{p}_{h^{e'}|Y^{e'}}(t|y) - \hat{p}_{h^e|Y^e}(t|y)\big)\mathrm{d}t\Big|$$

$$\leq \quad \int_{t \in \mathbb{R}^d} \big|\hat{p}_{h^{e'}|Y^{e'}}(t|y) - \hat{p}_{h^e|Y^e}(t|y)\big|\mathrm{d}t$$

$$\leq \quad \int_{|t| \leq r_2} \big|\hat{p}_{h^{e'}|Y^{e'}}(t|y) - \hat{p}_{h^e|Y^e}(t|y)\big|\mathrm{d}t$$

$$\quad + r_2^{-\alpha}\int_{|t| > r_2} \big|\hat{p}_{h^{e'}|Y^{e'}}(t|y) - \hat{p}_{h^e|Y^e}(t|y)\big||t|^\alpha\mathrm{d}t$$

$$\leq \quad \int_{|t| \leq r_2} \big|\hat{p}_{h^{e'}|Y^{e'}}(t|y) - \hat{p}_{h^e|Y^e}(t|y)\big|\mathrm{d}t + 2M_2 r_2^{-\alpha}.$$

Plugging the above upper bound into $I_1$,

$$I_1 \quad \leq \quad \int_{|h| \leq r_1}\int_{|t| \leq r_2} \big|\hat{p}_{h^{e'}|Y^{e'}}(t|y) - \hat{p}_{h^e|Y^e}(t|y)\big|\mathrm{d}t\mathrm{d}h + \int_{|h| \leq r_1} 2M_2 r_2^{-\alpha}\mathrm{d}h$$

$$\leq \quad \frac{\pi^{d/2}}{\Gamma(d/2 + 1)}r_1^d \times I_3 + \frac{2M_2\pi^{d/2}}{\Gamma(d/2 + 1)}r_1^d r_2^{-\alpha}$$

where

$$I_3 = \int_{|t| \le r_2} \big| \hat{p}_{h^{e'}|Y^{e'}}(t|y) - \hat{p}_{h^e|Y^e}(t|y) \big| \mathrm{d}t.$$

Note that $p_{h^e|Y^e}(t|y) \in L^1(\mathbb{R}^d) \cap L^2(\mathbb{R}^d)$. By the Projection theorem [6, 3],

$$\widehat{\mathcal{R}_e}(\beta, u) = \hat{p}_{h^e|Y^e}(u\beta|y), \quad u \in \mathbb{R}, \ \beta \in S^{d-1}, \tag{2}$$

where $\mathcal{R}_e(\beta, u)$ is the Radon transform of $p_{h^e|Y^e}(t|y)$:

$$\mathcal{R}_e(\beta, u) = \int_{h:\langle h, \beta \rangle = u} p_{h^e|Y^e}(h|y) \mathrm{d}h$$

and $\widehat{\mathcal{R}_e}(\beta, w)$ is the Fourier transform of $\mathcal{R}_e(\beta, u)$ with respect to $u$:

$$\widehat{\mathcal{R}_e}(\beta, w) = \int_{u \in \mathbb{R}} \exp(iuw) \mathcal{R}_e(\beta, u) du.$$

Thus we can rewrite the term $I_3$ as

$$\begin{aligned}
I_3 &= \int_{\beta \in S^{d-1}} \int_{|w| \in [0, r_2]} |w|^{d-1} \big| \widehat{\mathcal{R}_{e'}}(\beta, w) - \widehat{\mathcal{R}_e}(\beta, w) \big| \mathrm{d}w \mathrm{d}\beta \\
&\le r_2^{d-1} \int_{\beta \in S^{d-1}} \int_{|w| \in [0, r_2]} \big| \widehat{\mathcal{R}_{e'}}(\beta, w) - \widehat{\mathcal{R}_e}(\beta, w) \big| \mathrm{d}w \mathrm{d}s \beta \\
&\le r_2^{d-1} \int_{\beta \in S^{d-1}} \int_{|w| \in [0, r_2]} \int_{u \in \mathbb{R}} \big| \mathcal{R}_{e'}(\beta, u) - \mathcal{R}_e(\beta, u) \big| \mathrm{d}u \mathrm{d}w \mathrm{d}\beta.
\end{aligned}$$

Since the problem is $(s(\cdot), \mathcal{I}^{\mathrm{inf}}(h, \mathcal{E}_{avail}))$-learnable, and $\forall \beta \in \mathcal{S}^{d-1}$, the informativeness of feature $\beta^\top h$ is lower bounded by

$$\mathcal{I}(\beta^\top h, \mathcal{E}_{avail}) \ge \mathcal{I}^{\mathrm{inf}}(h, \mathcal{E}_{avail}),$$

we know that for any $\beta \in \mathcal{S}^{d-1}$,

$$\mathcal{V}(\beta^\top h, \mathcal{E}_{all}) \le s\big( \mathcal{V}(\beta^\top h, \mathcal{E}_{avail}) \big).$$

Therefore, we have

$$\mathcal{V}^{\mathrm{sup}}(h, \mathcal{E}_{all}) = \sup_{\beta \in \mathcal{S}_{d-1}} \mathcal{V}(\beta^\top h, \mathcal{E}_{all}) \le \sup_{\beta \in \mathcal{S}_{d-1}} s\big( \mathcal{V}(\beta^\top h, \mathcal{E}_{avail}) \big) = s\big( \mathcal{V}^{\mathrm{sup}}(h, \mathcal{E}_{avail}) \big).$$

Note that, for any given $\beta$, $\mathcal{R}_e(\beta, u)$ is the probability density of the projected feature $\beta^\top h$. So for any $e', e \in \mathcal{E}_{all}$,

$$\int_{u \in \mathbb{R}} \big| \mathcal{R}_{e'}(\beta, u) - \mathcal{R}_e(\beta, u) \big| \mathrm{d}u \le s(\mathcal{V}^{\mathrm{sup}}(h, \mathcal{E}_{avail})).$$

Therefore,

$$I_3 \le 2r_2^d \times \frac{\pi^{d/2}}{\Gamma(d/2 + 1)} \times s(\varepsilon).$$

Combining the result of $I_1$, $I_2$ and $I_3$, we have

$$I \le \frac{2\pi^d}{\Gamma^2(d/2+1)} r_1^d r_2^d s(\mathcal{V}^{\mathrm{sup}}(h, \mathcal{E}_{avail})) + \frac{2M_2 \pi^{d/2}}{\Gamma(d/2+1)} r_1^d r_2^{-\alpha} + 2M_1 r_1^{-\alpha}.$$

We take

$$r_1 = M_1^{\frac{1}{\alpha+d}} M_2^{-\frac{d}{(\alpha+d)^2}} s(\mathcal{V}^{\mathrm{sup}}(h, \mathcal{E}_{avail}))^{-\frac{\alpha}{(\alpha+d)^2}} \quad \text{and} \quad r_2 = M_2^{\frac{1}{\alpha+d}} s(\mathcal{V}^{\mathrm{sup}}(h, \mathcal{E}_{avail}))^{-\frac{1}{\alpha+d}}.$$

Hence

$$I \le \Big( \frac{2\pi^d}{\Gamma^2(d/2+1)} + \frac{2\pi^{d/2}}{\Gamma(d/2+1)} + 2 \Big) M_1^{\frac{d}{\alpha+d}} M_2^{\frac{\alpha d}{(\alpha+d)^2}} s(\mathcal{V}^{\mathrm{sup}}(h, \mathcal{E}_{avail}))^{\frac{\alpha^2}{(\alpha+d)^2}}.$$

The proof is finished. □

## 2.2 Proof of Theorem 4.2

**Theorem** *Consider any loss satisfying $\ell(\hat{y}, y) = \sum_{k=1}^{K} \ell_0(\hat{y}_k, y_k)$. Let the loss function $\ell_0(\cdot, \cdot)$ be bounded by $[0, C]$.*

*For any classifier with linear top model $g$, i.e.,*

$$f(x) = Ah(x) + b \quad with \quad A \in \mathbb{R}^{K \times d}, \ b \in \mathbb{R}^K,$$

*if $(\mathcal{E}_{avail}, \mathcal{E}_{all})$ is $\big(s(\cdot), \mathcal{I}^{inf}(h, \mathcal{E}_{avail})\big)$-learnable under $\Phi$ with Total Variation $\rho$, then we have*

$$\mathrm{err}(f) \leq O\Big(s\big(\mathcal{V}^{sup}(h, \mathcal{E}_{avail})\big)\Big). \tag{3}$$

*Here $O(\cdot)$ depends only on $d$ and $C$.*

*Proof.* For any $e \in \mathcal{E}_{avail}$ and $e' \in \mathcal{E}_{all}$, we know that $\mathbb{P}_Y(y) = \mathbb{P}_{Y^e}(y) = \mathbb{P}_{Y^{e'}}(y)$. Furthermore the generalization gap between $e$ and $e'$ is

$$
\begin{aligned}
& \mathbb{E}\big[\ell(f(X^{e'}), Y^{e'})\big] - \mathbb{E}\big[\ell(f(X^e), Y^e)\big] \\
= & \sum_{y=1}^{K} \mathbb{P}_Y(y)\Big(\mathbb{E}\big[\ell(f(X^{e'}), Y^{e'})\big|Y^{e'} = y\big] - \mathbb{E}\big[\ell(f(X^e), Y^e)\big|Y^e = y\big]\Big) \\
= & \sum_{y=1}^{K} \mathbb{P}_Y(y)\Big(\mathbb{E}\big[\sum_{j=1}^{K} \ell_0(f(X^{e'})_j, y_j)\big|Y^{e'} = y\big] - \mathbb{E}\big[\sum_{j=1}^{K} \ell_0(f(X^e)_j, y_j)\big|Y^e = y\big]\Big) \\
= & \sum_{y=1}^{K}\sum_{j=1}^{K} \mathbb{P}_Y(y)\Big(\mathbb{E}\big[\ell_0(f(X^{e'})_j, y_j)\big|Y^{e'} = y\big] - \mathbb{E}\big[\ell_0(f(X^e)_j, y_j)\big|Y^e = y\big]\Big),
\end{aligned}
$$

where $f(X^{e'})_j = A_j h(x) + b_j$. Here $A_j$ is the $j$-th row of the matrix $A$ and $b_j$ stands for the $j$-th element of the vector $b$. Then it suffices to uniformly bound

$$
\begin{aligned}
& \Big|\mathbb{E}\big[\ell_0(f(X^{e'})_j, y_j)\big|Y^{e'} = y\big] - \mathbb{E}\big[\ell_0(f(X^e)_j, y_j)\big|Y^e = y\big]\Big| \\
= & \Big|\int_{\mathbb{R}^d} \ell_0(A_j h + b_j, y)\big(p_{h^{e'}|Y^{e'}}(h|y) - p_{h^e|Y^e}(h|y)\big)\mathrm{d}h\Big|,
\end{aligned}
$$

where $h^e$ is the $d$-dimensional random vector $h(X^e)$. Let $t = A_j h + b_j$. Then,

$$
\begin{aligned}
& \Big|\int_{\mathbb{R}^d} \ell_0(A_j h + b_j, y)\big(p_{h^{e'}|Y^{e'}}(h|y) - p_{h^e|Y^e}(h|y)\big)\mathrm{d}h\Big| \\
= & \Big|\int_{t \in \mathbb{R}} \int_{\frac{A_j}{\|A_j\|_2}h + \frac{b_j}{\|A_j\|_2} = \frac{t}{\|A_j\|_2}} \ell_0(t, y)\big(p_{h^{e'}|Y^{e'}}(h|y) - p_{h^e|Y^e}(h|y)\big)\mathrm{d}h\mathrm{d}t\Big| \\
\leq & \ C \times \Big|\int_{t \in \mathbb{R}} \mathcal{R}_{e'}\big(\frac{A_j}{\|A_j\|_2}, \frac{t - b_j}{\|A_j\|_2}\big) - \mathcal{R}_e\big(\frac{A_j}{\|A_j\|_2}, \frac{t - b_j}{\|A_j\|_2}\big)\mathrm{d}t\Big| \\
\leq & \ O\big(s(\mathcal{V}^{sup}(h, \mathcal{E}_{avail}))\big).
\end{aligned}
$$

Hence

$$\mathbb{E}\big[\ell(f(X^{e'}), Y^{e'})\big] - \mathbb{E}\big[\ell(f(X^e), Y^e)\big] \leq O\big(s(\mathcal{V}^{sup}(h, \mathcal{E}_{avail}))\big).$$

$\square$

## 2.3 Proof of Theorem 4.3

**Theorem** *Consider* 0-1 *loss: $\ell(\hat{y}, y) = \mathbb{I}(\hat{y} \neq y)$. For any $\delta > 0$ and any expansion function satisfying 1) $s'_+(0) \triangleq \lim_{x \to 0+} \frac{s(x) - s(0)}{x} \in (1, +\infty)$; 2) exists $k > 1, t > 0$, s.t. $kx \leq s(x) < +\infty, x \in [0, t]$, there exists a constant $C_0$ and an OOD generalization problem $(\mathcal{E}_{avail}, \mathcal{E}_{all})$ that is $(s(\cdot), \delta)$-learnable under linear feature space $\Phi$ w.r.t symmetric KL-divergence $\rho$, s.t. $\forall \varepsilon \in [0, \frac{t}{2}]$,*

*the optimal classifier $f$ satisfying $\mathcal{V}^{sup}(h, \mathcal{E}_{avail}) = \varepsilon$ will have the OOD generalization error lower bounded by*

$$\text{err}(f) \geq C_0 \cdot s(\mathcal{V}^{sup}(h, \mathcal{E}_{avail})) \tag{4}$$

*Proof.* The expansion function $s(x)$ satisfies $kx \leq s(x) < +\infty$, $x \in [0, t]$. Construct an another function as:

$$\tilde{s}(x) = \begin{cases} kx & x \leq t \\ s(x) & x > t \end{cases}.$$

Clearly, $\tilde{s}(\cdot)$ is also an expansion function. According to Lemma 2.1, for $(\tilde{s}(x), \delta)$, there exists a constant $C_1 > 0$ and $(\mathcal{E}_{avail}, \mathcal{E}_{all})$ that is $(\tilde{s}(\cdot), \delta)$, s.t. for any $\mathcal{V}^{sup}(h, \mathcal{E}_{avail}) \leq \frac{t}{2}$, the optimal classifier $f$ satisfies

$$\text{err}(f) \geq C_1 \tilde{s}(\mathcal{V}^{sup}(h, \mathcal{E}_{avail})) = C_1 k_1 \mathcal{V}^{sup}(h, \mathcal{E}_{avail}).$$

Then it suffices to find a constant $C_0'$ such that

$$\mathcal{V}^{sup}(h, \mathcal{E}_{avail}) \geq C_0' s(\mathcal{V}^{sup}(h, \mathcal{E}_{avail})).$$

Notice that $s'_+(0) = M' \in (1, +\infty)$. Thus there exists $\delta$ such that $\forall x \in [0, \delta]$, $\frac{s(x)}{x} \leq 2M'$. In addition, $s(x) \leq M$, $x \in [0, t/2]$. Then, for any $x \geq \delta$, $\frac{x}{s(x)} \geq \frac{\delta}{M}$. Let $C_0' = \max\{\frac{\delta}{M}, \frac{1}{2M'}\}$. So, for any $x \in [0, t/2]$, $x \geq C_0' s(x)$. The proof is finished. $\qquad\square$

**Lemma 2.1** (lower bound for linear expansion function). *Consider* 0-1 *loss* $\ell(\hat{y}, y) = \mathbb{I}(\hat{y} \neq y)$. *For any linear expansion function* $s(x) = kx$, $x \in [0, t]$, $k \in (1, +\infty)$ *and any* $\delta > 0$, *there exists a constant* $C_1$ *and an OOD generalization problem* $(\mathcal{E}_{avail}, \mathcal{E}_{all})$ *that is* $(s(\cdot), \delta)$-*learnable under linear feature space* $\Phi$ *with symmetric KL-divergence* $\rho$, *s.t.* $\forall \varepsilon \leq \frac{t}{2}$, *the optimal classifier* $f$ *satisfying* $\mathcal{V}^{sup}(h, \mathcal{E}_{avail}) = \varepsilon$ *have* $\text{err}(f)$ *bounded by*

$$err(f) \geq C_1 \cdot s(\mathcal{V}^{sup}(h, \mathcal{E}_{avail})). \tag{5}$$

*Proof.* We construct $(\mathcal{E}_{avail}, \mathcal{E}_{all})$ as a binary classification task, where there are two domains in $\mathcal{E}_{avail}$, denoted as $\{1, 2\}$, and other two domains in $\mathcal{E}_{all} \setminus \mathcal{E}_{avail}$, denoted as $\{3, 4\}$. The dataset $(x, y)$ for domain $e \in \mathcal{E}_{all}$ is constructed as

$$y \sim \text{unif}\{-1, 1\}, \ z \sim \mathcal{N}(ry, 1), \ \eta^e \sim \mathcal{N}(a_e y, 1), \ x^e = \begin{pmatrix} z \\ \eta^e \end{pmatrix}.$$

Here we set

$$a_1 = -\sqrt{\frac{t}{2}}, \ a_2 = \sqrt{\frac{t}{2}}, \ a_3 = -\sqrt{\frac{kt}{2}}, \ a_4 = \sqrt{\frac{kt}{2}}, \ r = \sqrt{t}.$$

For any $\mathbf{w} = (w_1, w_2)^\top$, the distribution of $\phi^e = \mathbf{w}^\top x^e$ given $y$ is

$$\phi^e | y \sim \mathcal{N}\big(y(w_1 r + w_2 a_e), \|\mathbf{w}\|^2\big).$$

Now we calculate the variation of the feature. Notice that the symmetric KL divergence $\rho$ of two Gaussian distributions $\mathbb{P}_1 \sim N(\mu_1, \sigma^2)$ and $\mathbb{P}_2 \sim N(\mu_2, \sigma^2)$ is

$$\begin{aligned} \rho(\mathbb{P}_1, \mathbb{P}_2) &= \frac{1}{2} D_{KL}(\mathbb{P}_1 \| \mathbb{P}_2) + \frac{1}{2} D_{KL}(\mathbb{P}_2 \| \mathbb{P}_1) \\ &= \frac{1}{2} \frac{1}{\sigma^2} (\mu_1 - \mu_2)^2. \end{aligned}$$

Therefore, we have

$$\mathcal{V}(\phi^e, \mathcal{E}_{avail}) = \sup_{y \in \{-1,1\}} \frac{w_2^2 |a_1 - a_2|^2}{2\|\mathbf{w}\|_2^2} = \frac{tw_2^2}{\|\mathbf{w}\|_2^2} \leq t,$$

and

$$\mathcal{V}(\phi^e, \mathcal{E}_{all}) = \sup_{y \in \{-1,1\}} \sup_{e, e'} \frac{w_2^2 |a_e - a_{e'}|^2}{2\|\mathbf{w}\|_2^2} = k \frac{tw_2^2}{\|\mathbf{w}\|_2^2}.$$

Thus, for any $\phi \in \Phi$,

$$s(\mathcal{V}(\phi, \mathcal{E}_{avail})) = k \frac{tw_2^2}{\|\mathbf{w}\|_2^2} \geq \mathcal{V}(\phi, \mathcal{E}_{all}).$$

Therefore the OOD generalization problem $(\mathcal{E}_{avail}, \mathcal{E}_{all})$ that is $(s(\cdot), \delta)$-learnable under linear feature space $\Phi$ with symmetric KL-divergence $\rho$.

**Optimal Classifier** Now we consider $h(x) = (\phi_1(x), \ldots, \phi_d(x))^\top$ such that $\mathcal{V}^{\text{sup}}(h, \mathcal{E}_{avail}) = \varepsilon \leq \frac{t}{2}$, and see what the optimal classifier is like. Let $\mathbf{w}_i = (w_{i1}, w_{i2})^\top$ be the coefficients of $\phi_i$.

If for any $i \in [d]$, $w_{i2} = 0$. Then $s(\mathcal{V}^{\text{sup}}(\phi, \mathcal{E}_{avail})) = 0$ and for any $f$, $\text{err}(f) = 0$. So inequality 5 holds.

Now suppose there exists $i_0 \in [d]$ such that $w_{i_0 2} \neq 0$. Without loss of generality, we assume $i_0 = 1$ and $\|\mathbf{w}_i\| \neq 0$ for any $i \in [d]$. We then claim that $\forall i \in [d], \exists c_i \in \mathbb{R}, \mathbf{w}_i = c_i \mathbf{w}_1$. Otherwise, there exists a normalized vector $\beta \in \mathbb{R}^d$ such that $\beta^\top h(x) = c'(0,1)^\top x$, and we have $\mathcal{V}(\beta^\top h, \mathcal{E}_{avail}) = t$, which is contradictory to $\mathcal{V}^{\text{sup}}(h, \mathcal{E}_{avail}) \leq \frac{t}{2}$.

Since $\phi_i = c_i \phi_1$, it is obvious that under any loss function, the loss of optimal classifier on $h$ is the same as the optimal classifier on $\phi_1$. In the following, we shall focus on the optimal loss on $\phi_1$.

Without loss of generality, we further denote $\phi_1$ as $\phi(x) = (w_1, w_2)x$ and $w_1 > 0$. Since $\mathcal{V}^{\text{sup}}(h, \mathcal{E}_{avail}) = \varepsilon \leq t/2$, $|w_2| \leq w_1$. In addition, we have $r > |a_e|, e \in \{1, 2\}$. Therefore, $w_1 r + w_2 a_e > 0$, $\text{sign}(y(w_1 r + w_2 a_e)) = \text{sign}(y)$, and we can easily realize that for any $e \in \mathcal{E}_{avail}$, the optimal classifier is $f(x) = \text{sign}(\phi(x))$.

The loss of $f$ in $\mathcal{E}_{avail}$ is

$$
\begin{aligned}
\mathcal{L}(\mathcal{E}_{avail}, f) &= \max e \in \{1, 2\} \frac{1}{2}\Big[\mathbb{P}[f(x^e) < 0 | Y = 1] + \mathbb{P}[f(x) > 0 | Y = -1]\Big] \\
&= \max e \in \{1, 2\} \mathbb{P}[f(x^e) < 0 | Y = 1] \\
&= \max_{e \in \{1,2\}} \int_{-\infty}^0 \frac{1}{\sqrt{2\pi}\|\mathbf{w}\|} \exp\Big(-\frac{1}{2} \frac{(\phi - (w_1 r + w_2 a_e))}{\|\mathbf{w}\|^2}\Big) d\phi \\
&= \max_{e \in \{1,2\}} \int_{w_1 r + w_2 a_e}^{+\infty} \frac{1}{\sqrt{2\pi}\|\mathbf{w}\|} \exp\Big(-\frac{1}{2}\frac{\phi^2}{\|\mathbf{w}\|^2}\Big) d\phi \\
&= \int_{w_1 r - |w_2|\sqrt{\frac{t}{2}}}^{+\infty} \frac{1}{\sqrt{2\pi}\|\mathbf{w}\|} \exp\Big(-\frac{1}{2}\frac{\phi^2}{\|\mathbf{w}\|^2}\Big) d\phi \\
&= \int_{\hat{w}_1 r - |\hat{w}_2|\sqrt{\frac{t}{2}}}^{+\infty} \frac{1}{\sqrt{2\pi}} \exp\Big(-\frac{1}{2}\phi^2\Big) d\phi,
\end{aligned}
$$

where $\hat{w}_1 = w_1 / \|\mathbf{w}\|$ and $\hat{w}_2 = w_2 / \|\mathbf{w}\|$. Similarly, we have

$$
\mathcal{L}(\mathcal{E}_{all}, f) = \int_{\hat{w}_1 r - |\hat{w}_2|\sqrt{\frac{kt}{2}}}^{+\infty} \frac{1}{\sqrt{2\pi}} \exp\Big(-\frac{1}{2}\phi^2\Big) d\phi.
$$

Combined together, the OOD generalization error of the optimal classifier with $\mathcal{V}^{\text{sup}}(h, \mathcal{E}_{avail}) = \varepsilon$ is

$$
\begin{aligned}
\text{err}(f) &= \int_{\hat{w}_1 r - |\hat{w}_2|\sqrt{\frac{kt}{2}}}^{+\infty} \frac{1}{\sqrt{2\pi}} \exp\Big(-\frac{1}{2}\phi^2\Big) d\phi - \int_{\hat{w}_1 r - |\hat{w}_2|\sqrt{\frac{t}{2}}}^{+\infty} \frac{1}{\sqrt{2\pi}} \exp\Big(-\frac{1}{2}\phi^2\Big) d\phi \\
&= \int_{\hat{w}_1 r - |\hat{w}_2|\sqrt{\frac{kt}{2}}}^{\hat{w}_1 r - |\hat{w}_2|\sqrt{\frac{t}{2}}} \frac{1}{\sqrt{2\pi}} \exp\Big(-\frac{1}{2}\phi^2\Big) d\phi \\
&\geq C(\sqrt{k} - 1)\sqrt{\frac{t}{2}} |\hat{w}_2| \\
&\geq C(\sqrt{k} - 1)\sqrt{\frac{t}{2}} |\hat{w}_2|^2 \\
&= \frac{C(\sqrt{k} - 1)\sqrt{\frac{t}{2}}}{kt} s(\mathcal{V}^{\text{sup}}(h, \mathcal{E}_{avail})).
\end{aligned}
$$

We finish our proof by choosing $C_1 = \frac{C(\sqrt{k}-1)\sqrt{\frac{t}{2}}}{kt}$.

$\square$

# 3 Experiment on Colored MNIST

In this section, we conduct experiment on ColoredMNIST, a hand designed OOD dataset, to illustrate why validation accuracy fail to select a good model in OOD dataset.

## 3.1 Colored MNIST

The Colored MNIST [1] is a common-used synthetic dataset in OOD generalization problem. In the dataset, picture is labeled with $0$ or $1$, and it contains two color channels, one of which being $28 \times 28$ pixels gray scale image from MNIST [4] while the other being a zero matrix. Let the grayscale image and the colored image be $X$ and $\tilde{X}$ respectively, i.e., $\tilde{X} = [X, 0]^\top$ and $\tilde{X} = [0, X]^\top$ correspond to red and green image. Given a domain $e \in [0, 1]$, for an original image $X$ with the label $\hat{Y} = \mathbb{I}\{\text{digit} <= 4\}$, the data point in Colored MNIST is constructed with

$$Y = \begin{cases} \hat{Y} & \text{w.p. } 0.75 \\ 1 - \hat{Y} & \text{w.p. } 0.25 \end{cases}, \quad \tilde{X}^e = \begin{cases} [X, 0]^\top & \text{w.p. } e + (1 - 2e)Y \\ [0, X]^\top & \text{w.p. } e + (1 - 2e)(1 - Y) \end{cases} \tag{6}$$

According to (6), the digit shape is invariant over domains, and the color is varying but might be more informative than the digit shape in some domains. The difficulty of OOD generalization is that we need to avoid learning color, since in $e \in \mathcal{E}_{all}$ the relationship between $e$ and $y$ might be entirely reversed.

## 3.2 Learnability of Colored MNIST

As a warm-up, We first prove that for any $\delta$, Colored MNIST is a $(s(\cdot), \delta)$-learnable OOD problem under any feature space $\Phi$ with the total variation distance $\rho$, where

$$s(\varepsilon) = \frac{\max_{e,e' \in \mathcal{E}_{all}} |e - e'|}{\max_{e,e' \in \mathcal{E}_{avail}} |e - e'|} \varepsilon. \tag{7}$$

Here we assume the original dataset MNIST is generated from a distribution.

*Proof.* Denote $\phi(\tilde{X}^e)$ as $\phi^e$ and the density of $\phi(\tilde{X}^e)|Y^e = y$ as $p_{e,y}(x)$. In addition, denote the density of $\phi([X, 0]^\top)|Y^e = y$ as $p_{e,y}^1(x)$ and the density of $\phi([0, X]^\top)|Y^e = y$ as $p_{e,y}^2(x)$. Therefore, we have $\forall e, y$,

$$p_{e,y}(x) = [e + (1 - 2e)y]p_{e,y}^1(x) + [e + (1 - 2e)(1 - y)]p_{e,y}^2(x)$$

Since the distance $\rho(\cdot, \cdot)$ is total variation, we know that for any two domains $e, e'$,

$$\rho\big(\mathbb{P}(\phi^e|Y^e = y), \mathbb{P}(\phi^{e'}|Y^{e'} = y)\big)$$

$$= \frac{1}{2} \int \big|p_{e,y}(x) - p_{e',y}(x)\big| \mathrm{d}x$$

$$= \frac{1}{2} \int \Big|[e + (1 - 2e)y]p_{e,y}^1(x) + [e + (1 - 2e)(1 - y)]p_{e,y}^2(x)$$

$$\qquad - [e' + (1 - 2e')y]p_{e',y}^1(x) - [e' + (1 - 2e')(1 - y)]p_{e',y}^2(x)\Big| \mathrm{d}x$$

Notice that $X$ is invariant across domains. Thus for all $x, y$,

$$p_{e,y}^1(x) = p_{e',y}^1(x), p_{e,y}^2(x) = p_{e',y}^2(x).$$

We can omit the subscript $e$ and

$$\rho\big(\mathbb{P}(\phi^e|Y^e = y), \mathbb{P}(\phi^{e'}|Y^{e'} = y)\big)$$

$$= \frac{1}{2} \int \Big|(e - e')(1 - 2y)p_y^1(x) - (e - e')(2y - 1)p_y^2(x)\Big| \mathrm{d}x$$

$$= |e - e'| \int \Big|p_y^1(x) - p_y^2(x)\Big| \mathrm{d}x$$

$$= C|e - e'|,$$

where $C$ is a constant independent to $e, \delta$. By choosing $e, e'$ separately in $\mathcal{E}_{avail}$ and $\mathcal{E}_{all}$, we can derive the expansion function of Colored MNIST.

$\square$

Table 1: Algorithm specific hyperparmeter choice

| Algorithms | ERM | CORAL | GroupDRO | Mixup | IRM |
|---|---|---|---|---|---|
| Penalty | - | $\lambda$=1,0.1,0.01 | $\eta$=0.1,0.01 | $\alpha$=0.1,0.2 | iter=1000,$\lambda$=1,10 |
| lr | | 1e-4,5e-5 | | | |
| steps | | 2500,5000 | | | |

## 3.3 Validation Accuracy VS Out-of-distribution Accuracy in Colored MNIST

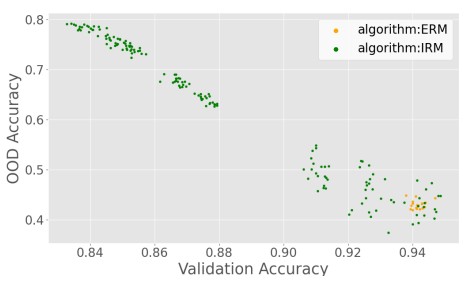

(a) Validation accuracy and OOD accuracy. They are *negative* correlated, i.e., high validation accuracy leads to low OOD accuracy.

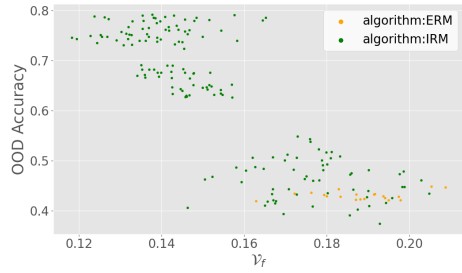

(b) Variation and OOD accuracy. They are *negative* correlated, i.e., low variation leads to high OOD accuracy.

Figure 2: Experiment Result on Colored MNIST.

We conduct experiments on the Colored MNIST dataset. As figure 2(a) shows, validation accuracy on Colored MNIST has a negative relation with OOD accuracy. Therefore, using validation accuracy as a metric to select will result in a poor OOD accuracy. By contrast, the correlation between variation OOD accuracy is also negative, meaning that the smaller the variation is, the higher the OOD accuracy will be.

## 4 Experiment Details

In this section we list our experiment details. We finish all of our experiment on 8 RTX3090 GPUs and 12 RTX2080 GPUs. It costs over 14,400 GPU hours.

**Architecture & Dataset** We use ResNet50 as our model architecture. The network except last linear and softmax layer is regarded as feature extractor $h(x)$ where the feature dimension $d = 2048$. We train our model on three real world OOD datasets (PACS [5], VLCS [9], OfficeHome [11]) by different algorithms and hyperparameters, and collect those models for selection procedure. Both datasets have 4 different environments. For each environment, we split it into 20% and 80% splits. The large part is used for training and OOD test. The small part is used for validation. We compare our criterion with validation criterion on each environment. We use Adam as our optimizer and weight decay is set to zero.

**Data Augmentation** Data augmentation is an important method for domain generalization problem. In our experiment, we follow same data augmentation setting in [2]. We first crops of random size and aspect ratio, resizing to 224 × 224 pixels, then we do random horizontal flips and color jitter. We also grayscale the image with 10% probability, and normalize image with the ImageNet channel means and standard deviations.

**Hyparameters & Algorithm** We search ERM [10] and four common OOD algorithms (Inter-domain Mixup [12], Group DRO [7], CORAL [8] and IRM [1]). Specific hyper-parameters are listed in Table 1. We train each setting for 5 times.

**Baseline** The performance of "Val" method is similar to another accuracy-based selection as is shown in [2]. We compare this method with ours.