# OpenReview forum: "Towards a Theoretical Framework of Out-of-Distribution Generalization"
_NeurIPS.cc/2021/Conference — NeurIPS 2021 Poster_

### Official Review · Reviewer_8ktt · 2021-07-15

**Rating:** 6
**Confidence:** 4

**Summary:**

This paper proves an error bound on domain generalization based on the a new concept of expansion function, which characterizes the extent that any useful feature's variance is amplified in the test domains over the training domains. The theory motivates a model selection criterion which has good empirical performance on benchmarks.

**Limitations And Societal Impact:**

No discussion of societal impact.

**Main Review:**

This paper studies the theoretical foundations of domain generalization, which is an important direction. The theoretical derivations are rigorous; presentation is clear, and the experiments support the theory.

However, there are a number of weaknesses:
1. The definition of expansion (3.4) is too strong, so the main results seem immediate from the assumptions.
2. The main bounds (Theorems 4.1, 4.2) should not hide factors in d.
3. The model selection criterion (equation 11) and the choice of $r_0$ should have more rigorous theoretical guarantees. Can the authors prove a theorem about the quality of this selection criterion?
4. It seems that every model selection criterion could be used to regularize the model during training. Based on this theory, one should regularize the conditional feature variance across training domains. However, algorithms based on this idea, e.g. conditional DANN, do not improve over ERM. This suggests that in realistic scenarios there are plenty of informative features with high OOD variances, so the main bounds are likely to be vacuous. In my opinion, a truly useful bound should include the complexity of feature extractor h.
5. In Algorithm 1 and Figure 1, why do the authors treat each dimension as a separate feature?
6. In the experiment, what happens when we take the sup over d dimensions instead of the average? Does that yield worse model selection performance?

**Time Spent Reviewing:**

5

---

> ### Author Response · Authors · 2021-08-07
> **Response to Reviewer 8ktt**
>
> Thank you for taking the time to review our work and providing very detailed and technical feedback. To our best understanding, some comments arise from misinterpreting our definitions. The variation, informativeness and expansion function is defined on each
> **one-dimensional** feature $\phi$ (Line 92-94), instead of on $d$-dimensional feature extractor $h$.
> We will highlight this issue in the reversion.
> Before the point-by-point response to your questions, we would like to highlight the merits of our assumptions: (i) Our assumptions are **mild** and widely used in the literature. (ii) Our assumptions are **practical** in the sense of efficient calculation for variation and informativeness. And we take this advantage in our empirical verifications.
>
>
>
> With this, here are our responses to your specific questions:
>
> > Q1. The definition of expansion (3.4) is too strong, so the main results seem immediate from the assumptions.
>
> > Q4. In my opinion, a truly useful bound should include the complexity of the feature extractor $h$.
>
> > Q5. In Algorithm 1 and Figure 1, why do the authors treat each dimension as a separate feature?
>
> **Ans:** We hope after our clarification of the assumption, these questions are already addressed: **Q1.** As mentioned above, we made mild and practical assumptions, and bounding the performance gap under this assumption is challenging and requires some advanced mathematical tools. **Q4.** We understand and agree that the complexity of $h$ should be included when we focus on bounding the distance of the high-dimensional feature distribution of $h$. Yet we took a different way of examining one-dimensional features.
> **Q5.** Just follow our definition.
>
> > Q2. The main bounds (Theorems 4.1, 4.2) should not hide factors in d.
>
> **Ans:** We were not intended to hide factors in $d$. We explicitly calculated the dependency on $d$ in Line 47 in Appendix but absorbed it into the big-O notation in the main body only because this term is too complicated and not the focus of the theorem. We are pleased to add it back if the reviewer suggests to.
>
> > Q3. The model selection criterion (equation 11) and the choice should have more rigorous theoretical guarantees. Can the authors prove a theorem about the quality of this selection criterion?
>
> **Ans:** We think that giving theoretical guarantees of a practical model selection method is difficult but interesting, and thank the reviewer for pointing out this future direction.
>
> > Q6. In the experiment, what happens when we take the sup over d dimensions instead of the average?
>
> **Ans:** In experiments, taking sup over $d$ makes the variation of different models indistinguishable. This is mainly because: in the theorem, we should provide guarantees for arbitrary (maybe adversarial) top models, so we should take the worst-case variation; but in practice, top models are learned and less adversarial, and taking the average can better capture the variation of different models.
>
> We sincerely hope our response can clarify your concerns and convince you to increase the score. Any further discussion is welcome.

---

> > ### Author Response · Authors · 2021-08-23
> > **Response to Reviewer 8ktt**
> >
> > Hello Reviewer 8ktt, we would be grateful if you can confirm whether our response has addressed your concerns and let us know if any issues remain. To recap our response,
> >
> > * We highlight that the definition of variation, informativeness and learnability is based on each one-dimensional feature rather than the $d$-dimensional feature extractor. The main theorems are not immediate results from the assumptions.
> >
> > * We clarify that the notation $O(1)$ in the manuscript is just for the simplicity of writing. The appendix gives the complete form of the upper bound of generalization.
> >
> > * We explain that taking the sup over $d$ dimensions is for the feasibility of the proof that gives a theoretical guarantee, while the average yields a better model selection performance in empirical studies.

---

> > > ### Comment · Reviewer_8ktt · 2021-09-02
> > > **Thanks for your response**
> > >
> > > Thanks for your clarifications, and I will raise my score to 6. I think the choice of one-dimensional features should be clarified in the revision.

---

> > > > ### Author Response · Authors · 2021-09-04
> > > > **Thank You for Increasing the Score**
> > > >
> > > > Thank you very much for increasing the score. We will clarify the choice of one-dimensional features in the revision.

---

### Official Review · Reviewer_MbMa · 2021-07-16

**Rating:** 8
**Confidence:** 3

**Summary:**

In this paper, the authors focus on understanding out-of-distribution (OOD) generalization through the variation of informative${}^{\small 1}$ features in training and unseen domains. Towards that, they introduce measures such as _variation_ and _informativeness_ as well as an _expansion function_, which are in turn used to introduce the concept of _learnability_. Providing generalization bounds, the authors show that OOD generalization can be characterized by the expansion function${}^{\small 2}$ under certain conditions on the _learnability_ property${}^{\small 3}$. Moreover, based on the findings, they propose a strategy for model selection (Algorithm 1). They further illustrate through experiments that their model selection strategy outperforms the ones selected using validation accuracy, and the constraint of informativeness is crucial to reason OOD generalization through variation.

${}^{\small 1}$defined over the training domain\
${}^{\small 2}$that forms a relationship between variations of informative features in training and unseen domains\
${}^{\small 3}$that informative features have limited variance in the unseen domain


**Ethical Concerns:**

I do not see any ethical issues within this paper.

**Limitations And Societal Impact:**

There is no discussion of societal impact but limitations are addressed.

**Main Review:**

- I very much enjoyed reading this paper. I do not have much knowledge about the prior work in regards to OOD generalization, but I find the paper very clear and well-written which is important especially for those who do not have a complete background on the topic _(clarity)_
- To my assessment, this work is original. That is, I am not aware of any prior work that forms a rigorous understanding of OOD generalization through variations of features. Moreover, the model selection strategy that is built upon this understanding is demonstrated to be effective through numerical experiments _(originality, significance, quality)_


**Time Spent Reviewing:**

3.5

---

> ### Author Response · Authors · 2021-08-07
> **Response to Reviewer MbMa**
>
> We sincerely thank you for your positive review!

---

### Official Review · Reviewer_eZ4t · 2021-07-16

**Rating:** 5
**Confidence:** 3

**Summary:**

This paper discusses a formalized approach to generalize to OOD data. It proposes assessing differences in the domains relying on distributional distances designed to account for variation in the feature distribution and informativeness of features. Generic bounds, and bound in the specific case of a linear classifier, are offered. Finally a practical heuristic algorithm is designed to perform model selection and experimentally evaluated.

**Limitations And Societal Impact:**

No assessment provided as the work is theoretical.

**Main Review:**

The work is interesting and it addresses a relevant problem. However I am not certain I completely understood or share some statements.
(line 123-125): "If a feature is informative for the classification task and invariant over \Epsilon_avail, then to enable OOD generalization from \Epsilon_avail to \Epsilon_all, it should be still invariant over \Epsilon_all" - I am not sure here about the meaning of "enable"; it seems to me that a feature that is informative for the classification task and invariant over \Epsilon_avail still does not tell us anything about generalization (as clearly shown by the colored MNIST example); if it is also invariant over \Epsilon_all, then it is a good feature for generalization. This seems to me to be better captured in line 145-148.

(line 128): "Expansion function": it seems to me that the expansion function could simply be swapped with a constant. I suppose the s() function is introduced to provided tighter bounds, is that so?

(line 155-156): "In addition, there are multiple choices of (s(), \delta) to make an OOD generalization problem learnable: larger \delta will filter out more features, and so s() can be smaller (flatter)": this may be clearer if stated mathematically, because larger \delta implies smaller or equal (<=) not necessarily smaller (<) expansion function (again thinking about the coloured MNIST example). In a sense the learnability of a problem seems to me not just something up to us tuning (s(), \delta), but a consequence of the problem itself. Also the wording "smaller" recalls the idea of expansion function as a constant. "with smaller derivative" may describe it better?

Experimental results are illustrative, although further validation of the theoretical results would have been interesting. Also more precise consideration on the computational complexity (beside line 297-298 that delegate KDE to a GPU) in relation to other algorithms would have been useful.

It would have been interesting to have a better connection with work done in the covariate shift adaptation (the setup of the problem seems to me to fit the covariate shift framework) as well as comparison with other classical theoretical work on assessing learning from different domains (Ben-David)

**Time Spent Reviewing:**

5

---

> ### Author Response · Authors · 2021-08-07
> **Response to Reviewer eZ4t**
>
>
> Thank you for the time spent on our work and for your extensive review.
>
> > Q1. The meaning of "If a feature is informative for the classification task and invariant over $\mathcal E_{avail}$, then to enable OOD generalization from  $\mathcal E_{avail}$ to  $\mathcal E_{all}$, it should be still invariant over  $\mathcal E_{all}$".
>
> **Ans:** We thank the reviewer for helping us clarify our statement. The meaning of this sentence is if a problem is learnable (enable OOD generalization), it should have the property that "if a feature is informative for the classification task and invariant over
> $\mathcal E_{avail}$, it should be still invariant over  $\mathcal E_{all}$". Then we introduce the idea of expansion function and learnability to describe this property. We will fix this sentence in the next version.
>
> > Q2. (line 128): I suppose the $s(\cdot)$ function is introduced to provide tighter bounds, is that so?
>
> **Ans:** Expansion Function should have $\lim_{x\to 0^+} s(x) = 0$, so a constant is not an expansion function to our definition. Of course, if $s(\cdot)$ is replaced by a sufficiently large constant $C$, the problem is $(C,\delta)$-learnable and our theorems hold. However, the expansion function is to describe the relationship between $\mathcal E_{avail}$ and $\mathcal E_{all}$, which cannot be achieved by constant function.  **Most importantly, when $s(\cdot)$ is a constant, the generalization gap will NOT goes to $0$ when the variation of the model (measured by $\mathcal V^{sup}(h,\mathcal E_{avail})$) goes to $0$.**
>
> > Q3. Problem of multiple choices of $(s(\cdot),\delta)$.
>
> **Ans:** Thanks for your suggestion on making the sentence stated mathematically. It should be: if the problem is $(s_1(\cdot),\delta_1)$ and $(s_2(\cdot),\delta_2)$-learnable where $\delta_1 \geq \delta_2$, then $\forall x \geq 0, s_1(x)\leq s_2(x)$. The statement "with smaller derivate" is more suitable, we will fix it in the next version. The learnability is a consequence of the dataset and the hypothesis set. In Appendix 3.2, we calculate the learnability of the Colored MNIST dataset, which may help the reviewer to better understand learnability.
>
> > Q4. Compare with Covariate Shift \& Computation Complexity
>
> **Ans:** Thanks for your suggestion and we will add some comparison in the next version. Briefly, the main difference is that our method considers one-dimensional feature distribution rather than $d$ dimension distribution distance (Ben-David, et al. 2010). This assumption is mild and more practical. We take this advantage in our empirical verifications.
>
> We hope that we have addressed your questions and comments adequately and sincerely hope that you can reconsider the score. Any further discussion is welcome.
>
>
> **Reference**
>
> Ben-David, S., Blitzer, J., Crammer, K., Kulesza, A., Pereira, F., & Vaughan, J. W. (2010). A theory of learning from different domains. Machine learning, 79(1), 151-175.

---

> > ### Author Response · Authors · 2021-08-23
> > **Response to Reviewer eZ4t**
> >
> > Hello Reviewer eZ4t, we would be grateful if you can confirm whether our response has addressed your concerns and let us know if any issues remain. To recap our response,
> >
> > * We clarify the definition of learnability and explain the meaning of the sentence: "If a feature is informative for the classification task and invariant over $\mathcal E_{avail}$, then to enable OOD generalization from  $\mathcal E_{avail}$ to $\mathcal E_{all}$, it should be still invariant over ".
> >
> > * We highlight the multi choices of $(s(\cdot), \delta)$ and explain that a constant function is not an expansion function.
> >
> > * We clarify the main difference between our method and the distribution shift framework: The definition of variation, informativeness and learnability is based on each one-dimensional feature.

---

> > > ### Author Response · Authors · 2021-09-04
> > > **Response**
> > >
> > > Hello Reviewer eZ4t, we hope that we have already addressed your concerns. Please let us know if you still have other concerns.

---

### Official Review · Reviewer_Ja6b · 2021-07-19

**Rating:** 6
**Confidence:** 3

**Summary:**

- They consider an OOD generalization setting, where we have a set of available domains we train on, E_avail, and a superset of test domains, E_all. They define L(f, E), the worst case loss of f on a set of domains E
- They define informativeness (how predictive a feature is for the label, worst case across domains) and variability (how much P(feature | y) varies across domains
- They define learnability: roughly speaking that all features that don't vary much in the available domains E_avail, also don't vary much in the E_all (for all features that are somewhat informative)
- Their main theorem roughly says that L(f, E_all) - L(f, E_avail) can be upper bounded by the variability of P(features | y) in E_all. By learnability this can be upper bounded by the variability of P(features | y) in E_avail. The notion of variability is the max TV distance between P(features | y) in different domains
- This inspires a model selection heuristic: look at the accuracy of a model on E_avail, but penalize models where the variability of the features is high, because this means the models may mess up OOD
- Their model selection strategy chooses better models across 3 standard datasets (PACS, Office-Home, VLCS), for all combinations of the OOD dataset.


**Limitations And Societal Impact:**

Looks alright!

**Main Review:**

After rebuttal: added comments below. I think the paper would be a nice addition to the conference.

---------------------------------------------------------------------------------------

Strengths
- The paper seems technically sound, and well-executed.
- I looked through the proof of Theorem 1 fairly carefully / line-by-line, and it looks correct. The only part I wasn't able to verify was the relation between Radon transform and the pdf, but they say this is from prior work.
- The experimental results are nice. The procedure looks simple, with not much potential for tuning, and seems to get improvements across 3 datasets, and combinations of OOD datasets.
- The idea of looking at variability of the features across domains, besides just accuracy, seems intuitive and reasonable, if you're interested in worst case accuracy across test domains
- The proof I read (Theorem 1) in the Appendix is very well written and easy to follow, more so than many other submissions. It looks like a complete piece of work.

Weaknesses
- There seem to be large gaps between the theory and practice.
- Average vs worst case: as one example, looking at variability makes sense to me if we are interested in **worst-case** error across test domains. For example, if f1 and f2 have the same worst-case accuracy in the available domains, but f1's features vary much less, I'd expect the worst-case accuracy on test domains to be better for f1. But it's not clear that it makes for *average* accuracy.
- The title is too general and strong. There are many theories for OOD generalization. I'd even view "A theory of learning from different domains", by Ben-David et al, as a theory for OOD generalization. It says if we train on a source domain, and the H-Delta-H divergence between source and target (OOD) is small, then you generalize well to the target (the model doesn't need to use any target data). Can you change it to something more specific? Something that conveys that you're studying feature variability and using it as a model selection strategy?

- I'm not sure if their assumption of learnability seems reasonable for a few reasons
1. It seems quite strong - to start with, could you give some reasonable examples where we would have a bound on learnability? For example, maybe suppose we have a distribution over domains, and we sample some domains randomly for E_avail. Under what conditions would we have learnability?
2. Let's look at the case where we have a distribution over domains. If variability was averaged across domains, then showing learnability holds seems like a standard concentration result. But in this case variability takes the max over domains, which doesn't concentrate as easily.
3. In addition, learnability requires the relation between the variability over E_all and E_avail to hold over **all** features, which could make the concentration much harder
4. My intuition also is that the harder problem is to show learnability. Once we show learnability, it isn't too surprising that the conclusion holds. If the features don't vary much in TV distance, then yeah the accuracy can't vary too much across domains…
5. I could also define other notions of learnability: for example a feature that gets high accuracy on all the available domains (that is the worst case accuracy over domains E_avail is high), gets high accuracy on all domains in E_all. In that case, my heuristic would be to pick a model that gets the best worst case loss on E_all. Why is this notion of learnability based on variability (more) interesting?

- For Theorem 1 and its proof: I'm not sure why you couldn't take the following simpler approach: Up to line 28 of the proof, you show that it suffices to bound C * I. I can be immediately bounded by taking the sup over e' \in E_all, e \in E_all, since this is a larger set since originally we had e' \in E_all, e \in E_avail, with E_avail \subset E_all
- But sup_{e, e' \in E_all} I = sup_{e, e' \in E_all} TV( P(h^e | y) , P(h^e' | y) ) \leq V_{TV}(h, E_all) \leq s(V_{TV}(h, E_avail))
- Here we would need (s, \delta) learnability, where \delta = I_{TV}(h, E_avail)
- The main change is we're using V_{TV}(h, E_avail) in the upper bound instead of V^sup as defined in the paper. It's a bit unclear why the additional complexity of defining V^sup, V^inf, and bounding various properties of the characteristic function was warranted

- They pick r_0 based on the standard deviations of accuracy and variability on the available datasets. This looks mathematically nice, but why is it a good choice?

- It could be nice for the theory to explain why using variability in model selection gives better models than just using the accuracy. I'm not quite able to piece together the existing theorems for this.

More comments:
- In equation 4, right, can't the characteristic function be complex? In that case, can't the integral be complex? If that's true then it may not make sense to upper bound by a scalar?
- In line 39 of the proof, in the supplement, the integral is over h, so should it be a dh instead of dv? If not, what is v?
- In the supplement, can you give more details on what models were chosen by your model selection procedure for each of the 12 cases in table 1?
- On line 297, it says "we design a parallel GPU kernel density estimation to speed up the whole process". I'm not sure I got any details of this (didn't find it in the Appendix), so I'm not sure how to factor this as a contribution.
- Maybe consider giving some background about the Radon transform in the Appendix, maybe even just copying over some of the basic properties from the lemma. This is very minor, but I think it can make things more readable

Overall, I like the idea of using feature variability to choose models, the experimental results are compelling, and they have some initial theoretical results. Good job!


**Time Spent Reviewing:**

6+1 hours

---

> ### Author Response · Authors · 2021-08-07
> **Response to Reviewer Ja6b**
>
> Thank you for your extensive and helpful comments and thorough review of the paper.
> We split our response into two parts. We start with your concerns on our assumptions and the simple proof you proposed.
>
> To our best understanding, some comments arise from misinterpreting our definitions.
> The variation, informativeness and expansion function are defined on each **one-dimensional** feature $\phi$ (Line 92-94), instead of on $d$-dimensional feature extractor $h$.
> We will highlight this issue in the reversion.
> Before the point-by-point response, we would like to highlight the merits of our assumptions: (i) Our assumptions are **mild** and widely used in the literature. (ii) Our assumptions are **practical** in the sense of efficient calculation for variation and informativeness. And we take this advantage in our empirical verifications.
>
> > Q4(1). It seems quite strong - to start with, could you give some reasonable examples where we would have a bound on learnability? For example, maybe suppose we have a distribution over domains, and we sample some domains randomly for $\mathcal E_{avail}$. Under what conditions would we have learnability?
>
> > Q5. For Theorem 1 and its proof: I'm not sure why you couldn't take the following simpler approach: Up to line 28 of the proof, you show that it suffices to bound $C \times I$. I can be immediately bounded by taking the sup over $e' \in \mathcal E_{all}$, $e \in \mathcal E_{all}$, since this is a larger set since originally we had $e' \in \mathcal E_{all}$, $e \in \mathcal E_{avail}$, with $\mathcal E_{avail} \subset \mathcal E_{all}.$
>
> > Q6. But $\sup_{e, e' \in \mathcal E_{all}} \mathcal I = \sup_{e, e' \in \mathcal E_{all}} TV( P(h^e | y), P(h^{e'} | y) ) \leq \mathcal V_{TV}(h, \mathcal E_{all}) \leq s(\mathcal V_{TV}(h, \mathcal E_{avail}))$.
>
> > Q7. Here we would need $(s, \delta)$ learnability, where $\delta = \mathcal I_{TV}(h, \mathcal E_{avail})$.
>
> **Ans:** With the above explanation, we hope that these questions are already resolved. We believe that our assumptions are not strong ** since it only requires one-dimensional feature's property.** Also, in your simpler approach, you mention $ \mathcal V_{TV}(h,\mathcal E_{all}) \leq s(\mathcal V_{TV}(h, \mathcal E_{avail}))$. Notice that we do not define variation ($\mathcal V$) for high dimensional feature $h$, so this step is incorrect. The major difficulty in the proof is exactly to recover some high dimensional property only from one dimension assumption.
>
>
> > Q4(2). Let's look at the case where we have a distribution over domains. If variability was averaged across domains, then showing learnability holds seems like a standard concentration result. But in this case, variability takes the max over domains, which doesn't concentrate as easily.
>
> **Ans:** In this work, we do not assume a meta distribution over domains. Both $\mathcal E_{avail}$ and $\mathcal E_{all}$ are fixed. Thanks for pointing out a potential future direction!
>
> > Q4(3). In addition, learnability requires the relation between the variability over $\mathcal E_{all}$ and $\mathcal E_{avail}$ to hold over **all** features, which could make the concentration much harder.
>
> **Ans:** We only require (1) one-dimensional and (2) **informative** features to satisfy variation relation. We believe these two constraints can make the assumptions quite practical, and Figure 1 for a real dataset (Office-Home) can illustrate this.
>
>
> > Q4(4). My intuition also is that the harder problem is to show learnability. Once we show learnability, it isn't too surprising that the conclusion holds. If the features don't vary much in TV distance, then yeah the accuracy can't vary too much across domains…
>
> > Q8. The main change is we're using $\mathcal V_{TV}(h, \mathcal E_{avail})$ in the upper bound instead of $\mathcal V^{sup}$ as defined in the paper. It's a bit unclear why the additional complexity of defining $\mathcal V^{sup}$, $\mathcal I^{inf}$, and bounding various properties of the characteristic function was warranted.
>
> **Ans:** If we just assume that each one-dimensional feature does not vary much in TV distance, the high dimension joint distribution can still do not overlap and thus the theorem breaks severely. We present a counterexample in Appendix 1. That's also the reason we need to consider $\mathcal V^{sup}$, i.e. the most varying one-dimensional feature in $h$. With this, we can guarantee the joint distribution distance is still small. We believe this proof is not trivial.
>
>
> > Q4(5). I could also define other notions of learnability: for example, a feature that gets high accuracy on all the available domains (that is the worst-case accuracy over domains $\mathcal E_{avail}$ is high), gets high accuracy on all domains in $\mathcal E_{all}$. In that case, my heuristic would be to pick a model that gets the best worst-case loss on $\mathcal E_{all}$. Why is this notion of learnability based on variability (more) interesting?
>
> **Ans:** Yes, we can also define learnability from the view of accuracy, which is straightforward. However, many classical OOD tasks like Colored MNIST are unlearnable under this definition. Our definition starts from the fact that "accuracy is not enough to judge OOD", and please see Appendix 3.1 and 3.2 for more discussions.
>
>
>
> **PART 2.** Response to the rest questions point-by-point.
>
> > Q1. There seem to be large gaps between the theory and practice.
>
> **Ans:** We are not very sure what gaps you mean. This work aims to build a practical theory that reflects some empirical intuition.
> We achieve this by: (i) define "informativeness, variation and learnability" on one-dimensional feature rather than on high-dimensional distribution (e.g., Ben-David et al. 2010). This is much weaker and practical, and we take this advantage in our experiments; (ii) derive both the upper and lower bound of the generalization error. These results show that the proposed framework can reflect the difficulty of a domain generalization task; (iii) propose a natural model selection criterion. This criterion comes from our theory and is demonstrated to be effective.
>
> > Q2. Average vs worst case: ... ... But it's not clear that it makes for average accuracy.
>
> **Ans:** In this work, we employ the variability to bound the worst-case accuracy. Indeed, for average accuracy in $\mathcal E_{all}$, our framework does not seem to be a completely suitable framework. However, the current Out-of-Distribution Generalization focuses mainly on the worst-case accuracy. we thank the reviewer for pointing out a possible future direction.
>
> > Q3. The title is too general and strong ...... Can you change it to something more specific? Something that conveys that you're studying feature variability and using it as a model selection strategy?
>
> **Ans:** Thank you for the suggestions. We would like to highlight the variability and change the current title in the revision.
>
> > Q9. They pick $r_0$ based on the standard deviations of accuracy and variability on the available datasets. This looks mathematically nice, but why is it a good choice?
>
> **Ans:** This idea comes from unifying the units (accuracy and variation). Since different tasks may have different levels of variation/accuracy, we think it's necessary to eliminate this gap.
>
> > Q10. It could be nice for the theory to explain why using variability in model selection gives better models than just using accuracy. I'm not quite able to piece together the existing theorems for this.
>
> **Ans:** Thank you for the comments. Intuitively, varying features may be ``risky'' in an unseen domain, and choosing features with high acc and low variation coincides with this intuition. From the theoretical aspect, small variation means a small generalization gap (our theorem). Therefore,
> $$
> \text{OOD accuracy} =  \text{ accuracy} (\mathcal E_{avail}) - \text{gap} \approx  \text{ accuracy}(\mathcal E_{avail}) - \text{variation},
> $$
> which leads to our model selection criterion.
>
>
> > Q11. In equation 4, right, can't the characteristic function be complex? In that case, can't the integral be complex? If that's true then it may not make sense to upper bound by a scalar?
>
> **Ans:** Thank you for pointing out this typo.
> Our proof requires
> $$\int_{t\in\mathbb{R}^d} |\hat p_{h^e|Y^e}(t|y)| |t|^\alpha \mathrm dt \leq M_2,$$
> where $|\cdot|$ stands for the Euclidean norm.
> We will fix it in the revision.
> In addition, the imaginary part of the characteristic function is an odd function.
> So the integral is real.
>
>
> > Q12. In line 39 of the proof, in the supplement, the integral is over $h$, so should it be a dh instead of $dv$? If not, what is $v$?
>
> **Ans:** Sorry for the typo. It should be $d h$. We will correct it in the revision.
>
> > Q13. In the supplement, can you give more details on what models were chosen by your model selection procedure for each of the 12 cases in table 1?
>
> > Q14. On line 297, it says "we design a parallel GPU kernel density estimation to speed up the whole process". I'm not sure I got any details of this (didn't find it in the Appendix), so I'm not sure how to factor this as a contribution.
>
> > Q15. Maybe consider giving some background about the Radon transform in the Appendix, maybe even just copying over some of the basic properties from the lemma. This is very minor, but I think it can make things more readable.
>
> **Ans:** We would like to answer the questions **Q13** - **Q15** together. Thank you for the constructive comments that make the manuscript more readable. In the revision, we will fix these problems.

---

> > ### Comment · Reviewer_Ja6b · 2021-09-02
> > **Response**
> >
> > Thanks for your response!
> > - Thanks for clarifying the definitions. I realized that the definitions are on 1-D, but in line 177 the paper takes the supremum over all projections.
> > - In the proposed simple proof, I mentioned "The main change is we're using V_{TV}(h, E_avail) in the upper bound instead of V^sup as defined in the paper.", so as you said I was proposing modifying the definition to hold over the d-dimensional features instead of the supremum over 1-d projections
> > - My main concern was "It's a bit unclear why the additional complexity of defining V^sup, V^inf, and bounding various properties of the characteristic function was warranted". I can believe that V^sup is weaker than over the d-dimensional feature space, but it's not clear to me why it makes a substantial practical difference in terms of modeling. Why is taking the sup over all directions mild, or reasonable?
> > - *Note that the theory looks correct so I'm not penalizing the paper for any of these*, and I do think it's above the bar! It's just that I can't give a strong accept if I don't really understand these modeling choices. I do agree the proof is not trivial.
> > - Maybe a tip for the future - if you say the assumptions are mild or practical, can you explain why? If the assumption is widely used, can you cite the assumptions?
> > - "There seem to be large gaps between the theory and practice." refers to the subsequent bullets, e.g. average vs worst case. Thanks for answering the questions.
> >
> > I think the paper would be a nice addition to the conference. Appendix 3.1 looks interesting, maybe discuss this in the main paper and explain it more (what's the result your bound gives for colored MNIST)

---

> > > ### Author Response · Authors · 2021-09-04
> > > **Response to Reviewer Ja6b**
> > >
> > > Thank you very much for appreciating the merits of our work and providing insightful comments.
> > >
> > > In the revision, we will clarify the assumptions of characteristic function and add citations, e.g., Cavalier, Laurent. "Efficient estimation of a density in a problem of tomography." Annals of Statistics (2000): 630-647. We will also modify Section 4 and Appendix 1 to highlight the necessity of $\mathcal V^{sup}.$ In Section 6，we will explain more about the efficient calculation of the one-dimensional assumption and $\mathcal V^{sup}.$

---

### Decision · Program_Chairs · 2021-09-28

**Decision:**

Accept (Poster)

**Comment:**

This paper investigates a variety of theoretical questions relating to out-of-distribution generalization. The reviewers found this paper to be very strong and well executed. The reviewers found the theoretical results to be sound, clearly explained, and compelling. Therefore, I recommend acceptance of this paper.

In the discussion, there are many suggestions for improvements to the story and the description of the motivation and intuition between some of the results and claims in the paper. I would encourage the authors to think about this, and other comments from the discussion, when writing the next version of the paper.

**Consistency Experiment:**

NeurIPS has a long history of experimentation. In 2014, NeurIPS ran an experiment in which 10% of submissions were reviewed by two independent committees to quantify the randomness in the review process. This year, we repeated a variant of this experiment to see how the quality of the review process has changed over time.  This paper was part of the experiment and was therefore assigned to two committees (consisting of reviewers, an Area Chair, and a Senior Area Chair) that reached independent decisions.  If both committees made the same recommendation, this recommendation was followed. If a single committee recommended acceptance, the paper was accepted (with the exception of a few cases in which the other committee identified what we considered a fatal flaw, e.g., an error in a key result).

This copy’s committee reached the following decision: **Accept (Poster)**

The other committee assigned to the paper recommended **Reject**.  You can find the other set of reviews, along with any follow up discussion with the authors here:
https://openreview.net/forum?id=kFJoj7zuDVi